# Research

ecology

beta-diversity decomposing, turnover and nestedness, multi-dimensional dissimilarity, opposite niche-based processes, Qinghai-Tibet plateau, montane system

**Authors for correspondence:**
Qisen Yang
e-mail: yangqs@ioz.ac.cn
Huijie Qiao
e-mail: huijieqiao@gmail.com

# A multi-faceted comparative perspective on elevational beta-diversity: the patterns and their causes

Yuanbao Du[1,†], Liqing Fan[2,3,4,†], Zhenghui Xu[5], Zhixin Wen[4], Tianlong Cai[6], Anderson Feijo[4], Junhua Hu[7], Fumin Lei[4], Qisen Yang[4] and Huijie Qiao[1]

[1]Key Laboratory of Animal Ecology and Conservation Biology, Institute of Zoology, Chinese Academy of Sciences, Beijing, People's Republic of China
[2]National Forest Ecosystem Observation & Research Station of Nyingchi Tibet, Institute of Plateau Ecology, Tibet Agriculture & Animal Husbandry University, Nyingchi, Tibet Autonomous Region, People's Republic of China
[3]Key Laboratory of Forest Ecology in Tibet Plateau (Tibet Agriculture & Animal Husbandry University), Ministry of Education, Nyingchi, Tibet Autonomous Region, People's Republic of China
[4]Key Laboratory of Zoological Systematics and Evolution, Institute of Zoology, Chinese Academy of Sciences, Beijing, People's Republic of China
[5]Key Laboratory of Forest Disaster Warning and Control in Yunnan Province, College of Biodiversity Conservation, Southwest Forestry University, Kunming, Yunnan Province, People's Republic of China
[6]School of Life Science, Westlake University, Hangzhou, Zhejiang Province, People's Republic of China
[7]Chengdu Institute of Biology, Chinese Academy of Sciences, Chengdu, Sichuan Province, People's Republic of China

YD, 0000-0003-1345-4127; AF, 0000-0002-4643-2293; JH, 0000-0001-9607-1863

The observed patterns and underlying mechanisms of elevational beta-diversity have been explored intensively, but multi-dimensional comparative studies remain scarce. Herein, across distinct beta-diversity components, dimensions and species groups, we designed a multi-faceted comparative framework aiming to reveal the general rules in the observed patterns and underlying causes of elevational beta-diversity. We have found that: first, the turnover process dominated altitudinal patterns of species beta-diversity ($\beta_{sim} > \beta_{sne}$), whereas the nestedness process appeared relatively more important for elevational trait dissimilarity ($\beta_{funcsim} < \beta_{funcsne}$); second, the taxonomic turnover was relative higher than its phylogenetic and functional analogues ($\beta_{sim} > \beta_{phylosim}/\beta_{funcsim}$), conversely, nestedness-resultant trait dissimilarity tended to be higher than the taxonomic and phylogenetic measures ($\beta_{funcsne} > \beta_{sne}/\beta_{phylosne}$); and third, as elevational distance increased, the contradicting dynamics of environmental filtering and limiting similarity have jointly led the elevational patterns of beta-diversity, especially at taxonomic dimension. Based on these findings, we infer that the species turnover among phylogenetic relatives sharing similar functional attributes appears to be the main cause of shaping the altitudinal patterns of multi-dimensional beta-diversity. Owing to the methodological limitation in the randomization approach, currently, it remains extremely challenging to distinguish the influence of the neutral process from the offset between opposing niche-based processes. Despite the complexities and uncertainties during species assembling, with a multi-dimensional comparative perspective, this work offers us several important commonalities of elevational beta-diversity dynamics.

# 1. Background

Encompassing a large number of endemic and threatened species within extremely limited spaces, montane regions are widely recognized as areas of high priority for conservation [1]. Before establishing any conservation measures and management schemes, one first needs to understand the mechanisms underlying

†Yuanbao Du and Liqing Fan equally contributed to this manuscript.

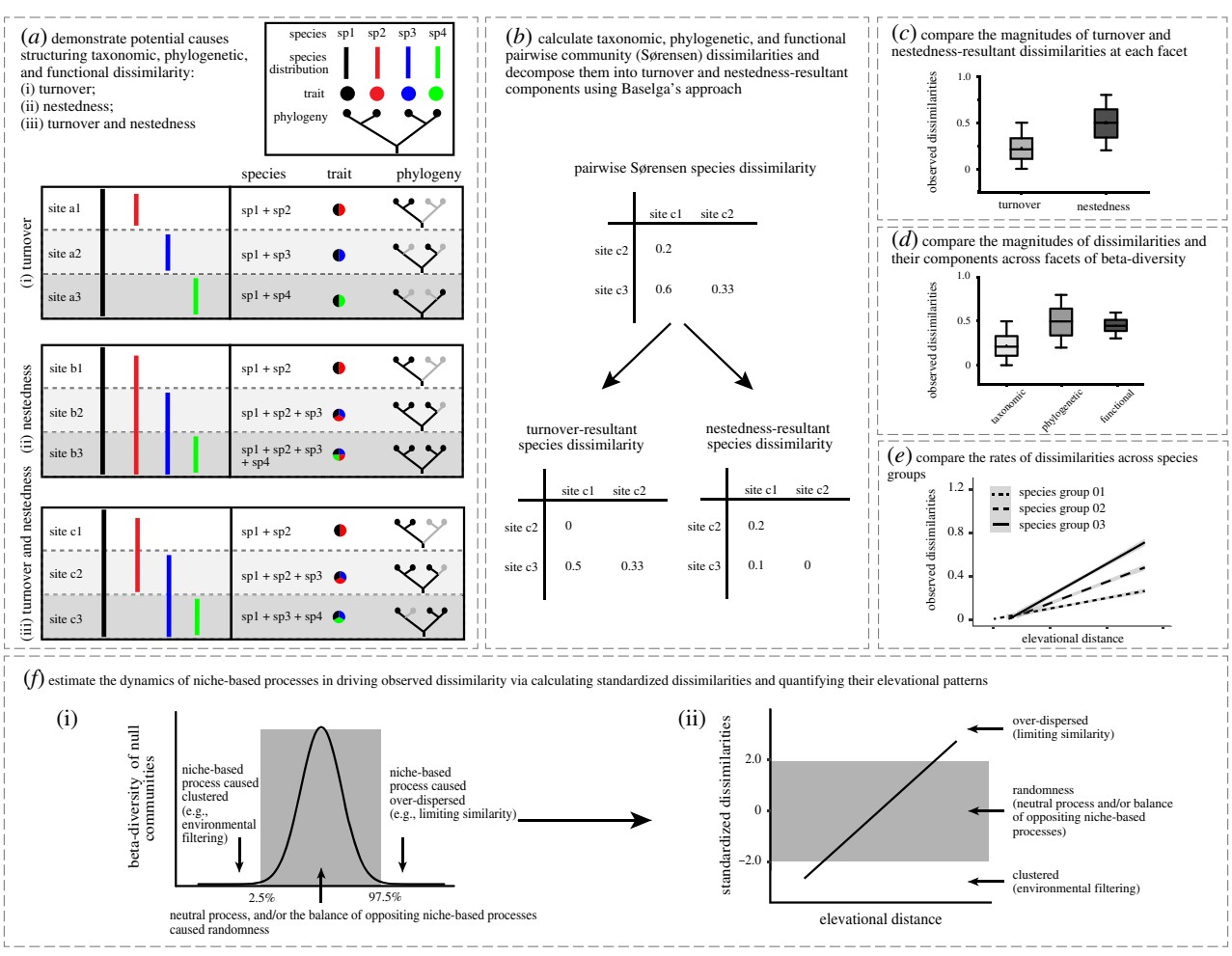

**Figure 1.** Conceptual figure outlines the analytical approach in this work. First, we demonstrate potential causes structuring taxonomic, phylogenetic and functional dissimilarity (*a*): (i) turnover; (ii) nestedness; and (iii) the joint of turnover and nestedness. Second, we calculate three dimensions of pairwise community dissimilarities and decompose into turnover and nestedness-resultant components following Baselga's approach (*b*). Third, we compare the magnitudes of turnover and nestedness-resultant dissimilarities at each facet of beta-diversity (*c*). Then, we compare the magnitudes of dissimilarities and their components across facets of beta-diversity (*d*). Across three species groups, we compare the rates of increase in dissimilarities (*e*). Lastly, we estimate the dynamics of opposite niche-based processes in driving observed dissimilarity via calculating standardized dissimilarities (*f*) (i) and quantifying their elevational patterns (*f*) (ii). (Online version in colour.)

community structure. Beta-diversity, which describes 'the extent of change in community composition among sites' [2], has become increasingly popular for understanding the drivers and maintenance of biodiversity [3,4]. Early beta-diversity studies primarily focused on taxonomic variation in community composition [2,5] and provided important insight into community assembly. However, as taxonomic classification alone cannot account for functional and evolutionary differentiation, mechanistic inferences on these grounds are increasingly questioned [6–8]. Phylogenetical and trait-based measures of beta-diversity, which estimates the phylogenetic and functional distance among communities, provide new perspectives by connecting local ecological and regional evolutionary processes [6,9,10]. Under the assumption of phylogenetic niche conservatism [7], if ecological niches of species group are phylogenetically conserved, the phylogenetic distance among species could be treated as a surrogate of interspecific dissimilarity at multiple niche dimensions [10–12]. However, ecological niches (frequently measured by ecological traits) are not always phylogenetically conserved [13,14] or, if conserved, cannot be fully accounted for with only a phylogeny [15–17]. These complexities demonstrate the importance of integrating taxonomic, phylogenetic and trait-based dimensions to understand the mechanisms determining community composition over space and time [16,18].

New approaches for beta-diversity decomposition offer opportunities to understand beta-diversity via the ecological causes of dissimilarity (e.g. spatial turnover and nestedness) [19]. Generally, the turnover component of community dissimilarity represents the contribution of replacement between distinct species, phylogenetic lineages or functional attributes [3] (figure 1*a*i), whereas nestedness-resultant dissimilarity results from ordered extinction or colonization along gradients [20–22] (figure 1*a*ii). Actually, these two antithetical processes are often mixed together (figure 1*a*iii), clearly showing the necessity of beta-diversity decomposing. Several approaches of decomposing community dissimilarity have seen significant study lately [3,23,24], and Baselga's approach based on turnover and nestedness processes is now widely accepted by community ecologists.

The relative importance of niche-based and neutral processes in structuring community has long been debated [25,26]. The neutral hypothesis emphasizes the importance of stochasticity and non-directional processes in driving community assembly [26], whereas the niche-based hypothesis emphasizes interspecific differentiation and non-random responses to environmental gradients [25]. Under the neutral hypothesis, the environment fitness of different organisms is assumed to be equivalent, of which neutral and stochastic processes are expected to be dominant in leading species

colonization, spatial distributions, and extinction [27]. That means, under a certain spatial and temporal extent, organisms with more effective dispersal ought to exhibit lower dissimilarity over space or time [28]. However, this prediction is not widely supported by empirical studies [29,30]. Well-known, heterogeneous biotic and abiotic interactions across space and time ought to affect the outcomes of neutral dispersal owing to species mutualisms or antagonisms [31–33]. Currently, the influence of both neutral and deterministic processes is widely accepted [12], but their relative importance and spatial dynamics typically vary across scales, environmental gradients and taxa [17,31–33].

Here, aiming to reveal the general patterns and underlying mechanisms of elevational beta-diversity, we designed a three-level comparative analysis across compositional components (turnover and nestedness), beta-diversity dimensions (species, phylogeny and traits) and taxonomic groups (passerines, rodents and ants) along a Tibetan mountain slope. We decomposed species, phylogenetic and trait-based dissimilarity of three animal groups (figure 1b). For the first level of comparison, we examined the magnitude of dissimilarity related to turnover and nestedness processes, and repeated the comparison at each dimension of beta-diversity in the three animal groups (figure 1c). For the second level of comparison, aiming to assess the magnitude of dissimilarity across the three beta-diversity dimensions, we performed inter-dimensional comparison on the dissimilarity of turnover, nestedness and their total contributions, respectively (figure 1d). Further, we attempted to examine the linear relationships of elevational beta-diversity and their best environmental predictors. If the observed beta-diversity exhibited a significant ($p \leq 0.05$) linear pattern with elevational distance, we then conducted the third-step of comparison by examining the slopes of linear models across the three animal groups (figure 1e). Predictively, we proposed a series of hypotheses (H0) and alternatives (H1) to illustrate potential scenarios and relevant mechanisms in the three-level comparisons (table 1). Lastly, by applying null-model procedures and linear regressions, we examined the dynamics of different niche-based processes (e.g. environmental filtering and negative competition exclusion) in driving elevational beta-diversity (figure 1f).

# 2. Materials and methods

## (a) Study location and data collection

This work was conducted on the eastern slope of Mt Segrila (29°10′–30°15′ N, 93°12′–95°35′ E) [39] (electronic supplementary material, figure S1), which is located in eastern Nyingchi City, Tibet, China. Located between the Nyainqêntanglha Mountains and the Himalayas, the elevational range of Mt Segrila reaches approximately 3400 m (valley base is approximately 1900 m; mountain summit exceeds 5300 m) [40]. This region has a monsoonal climate, and warm-temperate, needleleaf and broadleaf mixed forests are typical in the low valley zone (1900–2700 m). As elevation increases, the temperate needleleaf forest zone (2700–3300 m), sub-alpine cold-temperate needleleaf forest zone (3300–4200 m), alpine cold-temperate shrub and meadow zone (4200–4500 m) and alpine tundra and desert zone (4500–5300 m) successively dominate on the eastern slope of Mt Segrila [40].

As the primary and secondary consumers and the most important food resources to senior consumers, birds, small mammals and invertebrates, respectively, play important roles in energy transmission and material circulation in various types of biocenoses and ecosystems. Considering our research aims and experimental feasibility, we chose passerines (Passeriformes), rodents (Rodentia) and ants (Formicidae) to represent these fauna groups. Within an approximate 2500 m altitudinal range along the eastern slope of Mt Segrila, we performed field surveys for these three animal groups. The compositional information of passerines was collected from 4 to 10 times' field observations of 17 sampling transects (800 m–2300 m) dating from June 2018/2019 to June 2020. The passerine dataset contained 116 species of 64 genera and 34 families. Field survey on rodents included snap-trapping and field observations. Snap-trappings were conducted from nine sampling sites with two trapping surveys during the early (March–July) and late (July–September) wet seasons in 2014. We also conducted three field observations as snap-trapping, of which the first and second were carried out as snap-trapping and the third field observation was conducted in August 2018. In total, we have collected 14 rodent species, in which two species (*Sciurotamias davidianus* and *Marmota himalayana*) were collected from field observations only. We collected 18 ant species (including three unidentified species) from 11 sampling sites during July–August 2009. As we found no ant in the highest sampling sites (4548 m), they were judged absent there. More detailed descriptions of field collections are given in the appendices (electronic supplementary material, Appendix S01). The datasets of raw presence–absence species composition are available at Dryad (doi:10.5061/dryad.mw6m905wf) [41].

## (b) Phylogeny

Phylogenies of rodents and ants used in phylogenetic beta-diversity analyses were reconstructed using published DNA sequences from GenBank (http://www.ncbi.nlm.nih.gov/genbank/). We inferred the rodent phylogeny using four mitochondrial DNA genes (Cytb, CoI, 12s-rRNA and 16s-rRNA) and three nuclear DNA genes (IRBP, GHR, and RAG1) (electronic supplementary material, figure S2) [15]. To reconstruct the ant phylogeny, we used two mitochondrial DNA genes (CoI and 28s-rRNA). Phylogenetic relationships for ant and rodent species were separately estimated through Bayesian inference using MRBAYES (version 3.2.5) [42]. Parameter settings and other detailed information in phylogenetic reconstruction can be found in Du *et al.* [15]. Phylogenetic relationships for passerines were obtained using the online phylogeny tool on the website BirdTree (https://birdtree.org/) [43]. This tool provided a simple way of producing distributions of trees with subsets of bird taxa. This approach followed the same structure and taxonomy as Jetz *et al.* [43], which has been widely used in recent integrative analyses. We chose the default setting 'Ericson All Species: a set of 10 000 trees with 9993 operational taxonomic units each' to generate 100 trees that were used in subsequent phylogenetic analyses for passerines. The phylogenetic information for passerines, rodents and ants is available in the electronic supplementary material, Appendix S02.

## (c) Functional traits: size-related morphological attributes

The term 'functional trait' is a definition of the measurable function relating to an organism's niche. By measuring the functional aspects of diversity that potentially affect community assembly [44], 'functional diversity' describes the extent of functional differences among the species in a community [44–46]. Aiming to conduct comparative analyses across three animal taxa, size-related morphological attributes were used to quantify trait-based beta-diversity. Body size is one of the most important characters affecting animal interspecific competition and resource access [47–49], and size-related morphological characters have been frequently applied in comparative analyses of the functional diversity of the animal community [31,32,50]. For rodents, we

**Table 1.** Potential patterns and relevant mechanisms in the three-level comparisons across components (turnover and nestedness), dimensions (species, phylogeny and traits) and species groups (passerines, rodents and ants). (Non-significant differences could result from the interactions between opposite processes and are not presented.)

| comparison levels | hypothesis (H0) and potential mechanisms | alternative hypothesis (H1) and potential mechanisms | reference |
|---|---|---|---|
| 1. within dimension | $\beta_{sim} > \beta_{sne}$: species dissimilarity is driven by species replacement resulting from different environment or strong geographical barrier | $\beta_{sim} < \beta_{sne}$: species dissimilarity is driven by neutral dispersal and stochastic extinction/colonization | [3] |
| | $\beta_{phylosim} > \beta_{phylosne}$: there are strong historical isolation and stable local adaptations for major phylogenetic lineages | $\beta_{phylosim} < \beta_{phylosne}$: there are frequent historical up-down connections and labile local adaptation | [3,6] |
| | $\beta_{funcsim} > \beta_{funcsne}$: stepwised environmental filtering along elevational gradient acts on highly labile functional attributes | $\beta_{funcsim} < \beta_{funcsne}$: increasingly environmental filtering along elevational gradient acts on the trait of weak lability | [34] |
| 2. across dimensions | $\beta_{sim} > \beta_{phylosim}/\beta_{sne} > \beta_{phylosne}/\beta_{sor} > \beta_{phylosor}$: species turnover/loss/gain occurs more likely among phylogenetic relatives | $\beta_{sim} < \beta_{phylosim}/\beta_{sne} < \beta_{phylosne}/\beta_{sor} < \beta_{phylosor}$: species turnover/loss/gain frequently happens among distant-related species | [35,36] |
| | $\beta_{sim} > \beta_{funcsim}/\beta_{sne} > \beta_{funcsne}/\beta_{sor} > \beta_{funcsor}$: species turnover/loss/gain frequently occurs among species of similar ecological performance | $\beta_{sim} < \beta_{funcsim}/\beta_{sne} < \beta_{funcsne}/\beta_{sor} < \beta_{funcsor}$: species turnover/loss/gain frequently occurs among species with different ecological performance | [35,36] |
| | $\beta_{phylosim} > \beta_{funcsim}/\beta_{phylosne} > \beta_{funcsne}/\beta_{phylosor} > \beta_{funcsor}$: the traits mediating deterministic and stochastic processes are conserved on phylogeny | $\beta_{phylosim} < \beta_{funcsim}/\beta_{phylosne} < \beta_{funcsne}/\beta_{phylosor} < \beta_{funcsor}$: the traits mediating deterministic and stochastic processes are convergent on phylogeny | [35,36] |
| 3. across species groups | $\beta_{ant} > \beta_{rodent} > \beta_{passerine}$ (species): lower dispersal efficiency produces higher beta-diversity; invertebrates are more sensitive to the environmental variations than homothermal mammals and birds | unpredictable: different species groups experience different efficient time of dispersal and depend on distinct biotic and abiotic factors; the effect of species pool | [29,32,36] |
| | $\beta_{ant} > \beta_{rodent}/\beta_{passerine}$ (phylogeny): stronger historical isolation will produce a higher rate of phylogenetic dissimilarity. Historical connections in rodents and passerines are more frequent than ants during the uplift of the Qinghai-Tibet Plateau and glacial-interglacial oscillation | unpredictable: different species groups experience distinct evolutionary process at inconsistent temporal scales | [6,37] |
| | $\beta_{ant} > \beta_{rodent}/\beta_{passerine}$ (trait): higher functional stability and higher environmental persistence produce lower trait beta-diversity | unpredictable: local environment adaptations of different species groups rely on a different organism–environment relationship | [38] |

characterized size using the mean of each measure (body weight, head-body length, tail length, ear length and hind-foot length) of at least eight adult specimens (four males and four females) for each species [15]. Similarly, six (body weight, body size, bill length, wing length, tail length and plantar) and two (maximum and minimum head-body length of worker-priests) size-related morphological traits were used to represent the size in passerines and ants, respectively. The head-body length of worker-priests indicates the total natural length of the head, thoracic segments and abdominal segments. The measurements were obtained from measuring specimens (rodents), historical records and the literature (passerines and ants). We performed principal component analysis with the transformed (scaled and log-transformed) morphological attributes. As the first two components account for 70–100% of the total variation, they were used to calculate sized-related trait beta-diversity in this work (rodents: electronic supplementary material, figure S2; ants: figure S3). We also assessed the degree of a phylogenetic signal of each size-related morphological attribute using the $K$ statistic proposed by Blomberg et al. [51]. The measurements of size-related morphologies and phylogenetic signal detection are available in the electronic supplementary material, Appendix S03.

## (d) Environmental variables

In order to assess the dependence of elevational beta-diversity on the local environment, we conducted best predictor selection on five climatic and MODIS factors: annual mean temperature (AMT), annual precipitation (AP), annual mean humidity (AMH), net primary production (NPP) and potential evapotranspiration (PET). AMT, AP and AMH were extracted from the climate records of eight auto local weather stations (Pailong, Nichi, Bingzhan, Lulang, 114zhan, Shengtaizhan, 113zhan, and Shanding) (dating from January 2007 to February 2008) on Mt Segrila (data shown in the electronic supplementary material, Appendix S04) [52]. The three climatic factors were estimated for each sampling site using a linear estimation based on the records of the two nearest weather stations. NPP and PET were extracted from high resolution (1 * 1 km) MODIS products accessed from the office website of Land Processes Distributed Active Archive Center (https://lpdaac.usgs.gov/) (accessed in July 2016) [53,54]. A layer mosaic was performed in ENVI (v. 4.7) [55]. Projection transformation and data extraction were carried out with ArcGIS (v. 10.0) [56].

## (e) Observed and standardized beta-diversities

In this work, the terms 'community' or 'assemblage' mean focal species occurring in each sampling site. The observed and standardized taxonomic, phylogenetic and trait beta-diversity across communities were calculated based on raw presence–absence community data using the incidence-based pairwise Sørensen dissimilarity index [38] and its phylogenetic and trait-based analogues [9,34]. Trait beta-diversity measures were calculated using the convex hull approach proposed by Villéger et al. [34]. Using an additive decomposing approach [3], pairwise beta-diversities were decomposed into turnover and nestedness-resultant dissimilarity components. Under a null model, the standardized measures were calculated for all observed beta-diversities. Owing to the methodological limitation of the convex hull approach, observed and standardized trait beta-diversity measures as well as the associated analyses were only calculated and assessed for assemblages containing three or more species. Detailed information for calculating observed and standardized beta-diversities are provided in the electronic supplementary material, Appendix S05.

## (f) Statistical analyses

The comparisons between turnover and nestedness-resultant dissimilarities, and the dissimilarities across three dimensions of beta-diversity were examined using a two-sided Wilcoxon rank-sum test [57]. The elevational patterns of observed dissimilarity and their components along elevational distances were determined via linear regressions. The Wilcoxon rank-sum test and linear fitting model for trait beta-diversities were only assessed among subset assemblages containing three or more species. Across species groups, we compared the rate of increase in observed dissimilarity with elevational distance by examining the slopes of linear regressions (if it is significant, $p \leq 0.05$). As the quantification of trait beta-diversity was discordant with each other, the comparisons across animal groups mainly focused on species and phylogenetic dimensions. According to the least Akaike information criterion, the selections of the best predictive variables for each observed beta-diversity measure were performed using a forward selection procedure. Using linear regressions, we examined the elevational patterns of standardized beta-diversity, with the aim to assess the dynamics of different deterministic processes as elevational distance increased.

All statistical analyses were performed in R (v. 3.5.3) [58] using the packages 'leaps' [59], 'foreach' [60], 'fBasics' [61], 'ape' [62], 'ecodist' [63], 'picante' [64], 'FD' [65], 'vegan' [66] and 'betapart' [67]. Box plots and scatter plots were generated using 'ggplot2'

[68]. The silhouette images of passerine, rodent and ant were freely obtained from PhyloPic (http://phylopic.org/).

# 3. Results

## (a) Three-level's comparison on observed beta-diversity

### (i) Comparisons between turnover and nestedness processes

According to the results of the Wilcoxon test, the dissimilarity of species turnover was relatively higher than that of nestedness ($\beta_{sim} > \beta_{sne}$), and this was consistent across the three animal groups (passerines: $|z| = 6.411$, $p < 0.001$; rodents: $|z| = 3.243$, $p = 0.001$; ants: $|z| = 2.382$, $p = 0.017$). On the contrary, at trait dimension, the nestedness-resultant dissimilarity was significantly ($p \leq 0.05$) or nearly significantly ($0.05 < p \leq 0.1$) higher than that of turnover (passerines: $|z| = 8.268$, $p < 0.001$; rodents: $|z| = 2.857$, $p = 0.001$; ants: $|z| = 1.887$, $p = 0.059$). The comparisons at phylogenetic dimension were inconsistent across the three animal groups (figure 2; electronic supplementary material, table S1).

### (ii) Comparisons across diversity dimensions

The comparisons of turnover ($\beta_{sim}$, $\beta_{phylosim}$, and $\beta_{funcsim}$) and nestedness-resultant dissimilarity ($\beta_{sne}$, $\beta_{phylosne}$ and $\beta_{funcsne}$) across beta-diversity dimensions exhibited consistent patterns across the three animal groups: species turnover was relatively higher than their phylogenetic and trait analogues ($\beta_{sim} > \beta_{phylosim}/\beta_{funcsim}$), whereas nestedness-resultant dissimilarities at trait dimension tended to be higher than their species and phylogenetic analogues ($\beta_{funcsne} > \beta_{phylosne}/\beta_{sne}$) (figure 2; electronic supplementary material, table S1). Owing to the uneven joint effects of turnover and nestedness processes, the results of comparative analyses on total dissimilarities were inconsistent across the three species groups (figure 2; electronic supplementary material, table S1).

### (iii) Comparisons across species groups: the rate of the increase in species and phylogenetic dissimilarity with increasing elevational distance

Across animal groups, we mainly focused on the comparisons at species and phylogenetic dimensions. According to the results of linear regressions, $\beta_{sim}$, $\beta_{sor}$ and $\beta_{phylosor}$ of the three animal groups consistently exhibited significant ($p < 0.05$) monotonic linear patterns, but not for $\beta_{sne}$ of passerines, $\beta_{sne}$ and $\beta_{phylosne}$ of rodents, and $\beta_{phylosim}$ and $\beta_{sne}$ of ants (electronic supplementary material, figure S4 and table S2). By comparing the slopes ($S$) of significant linear fitting models, we found that species turnover and total species dissimilarity of rodents ($S$-$\beta_{sim} = 4.66 \times 10^{-04}$ and $S$-$\beta_{sor} = 4.02 \times 10^{-04}$) increased more quickly than those of passerines ($S$-$\beta_{sim} = 2.05 \times 10^{-04}$ and $S$-$\beta_{sor} = 2.00 \times 10^{-04}$) and ants ($S$-$\beta_{sim} = 2.00 \times 10^{-04}$ and $S$-$\beta_{sor} = 2.50 \times 10^{-04}$). At the phylogenetic dimension, the rate of increase in the phylogenetic dissimilarity of ants ($S$-$\beta_{phylosor} = 2.33 \times 10^{-04}$) was slightly higher than that of rodents ($S$-$\beta_{phylosor} = 2.08 \times 10^{-04}$) and passerines ($S$-$\beta_{phylosor} = 1.50 \times 10^{-04}$) (electronic supplementary material, figure S4 and table S2).

## (b) Best environmental predictors

According to the results of forward model selections for observed measures of beta-diversity, the best environmental predictors

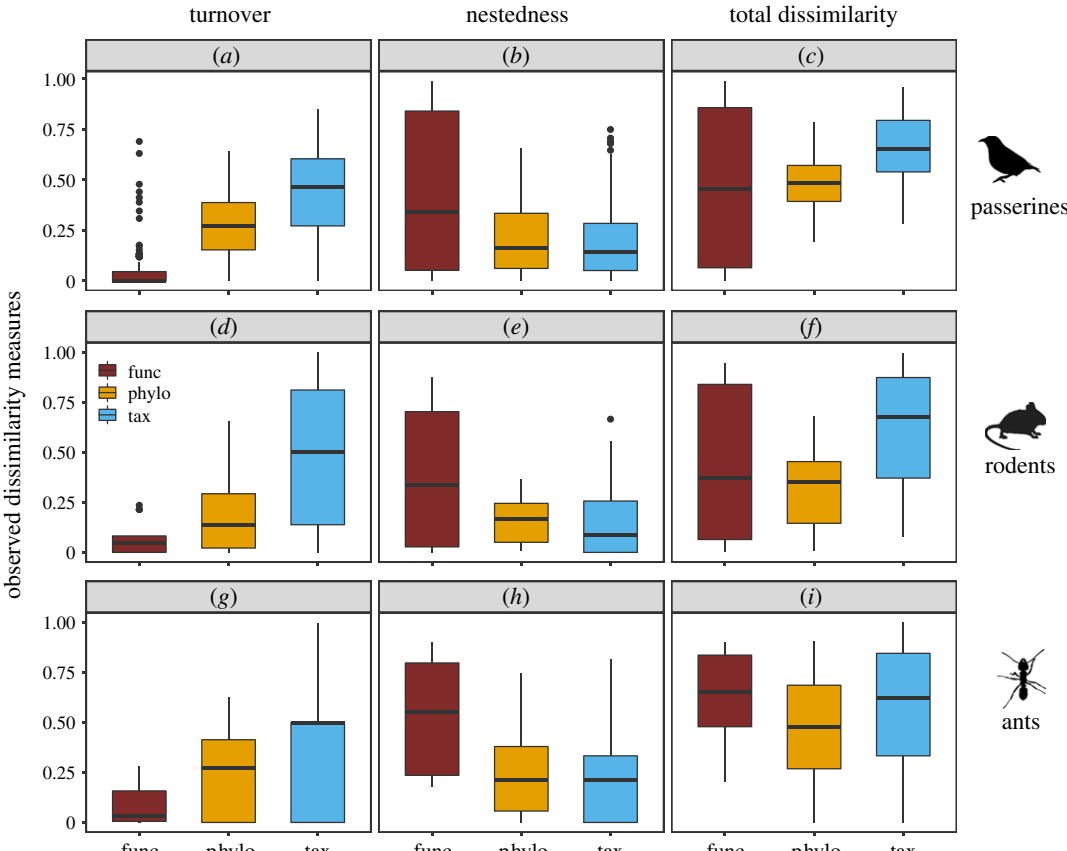

**Figure 2.** The dissimilarities driven by turnover (*a*, *d* and *g*), nestedness (*b*, *e* and *h*) and their combinations (*c*, *f* and *i*) at taxonomic ('tax', light blue (light grey)), phylogenetic ('phylo', orange (grey)) and functional ('func', dark red (dark grey)) dimensions in passerines (*a*–*c*), rodents (*d*–*f*) and ants (*g*–*i*). Observed functional beta-diversity measures of rodents and ants were calculated using subset assemblages containing three or more species. (Online version in colour.)

were inconsistent across beta-diversity components (turnover and nestedness), dimensions (species, phylogeny and trait) and animal groups (passerines, rodents and ants) (electronic supplementary material, table S3). Among five environmental predictors, NPP significantly contributed to the elevational patterns of 17 observed beta-diversity measures, followed by MAT (14), PET (10) and AP (six). By comparison, AMH significantly contributed to only three observed beta-diversity measures (electronic supplementary material, table S3).

## (c) Standard pairwised beta-diversity measures: the dynamics of opposite deterministic processes

According to the results of the linear regressions, the standardized taxonomic measures (SES . $\beta_{sim}$, SES . $\beta_{sne}$ and SES . $\beta_{sor}$) of the three animal groups consistently exhibited significant ($p \leq 0.05$) or nearly significant ($0.05 < p \leq 0.1$) linear patterns as elevational distance increased. Most of the standardized phylogenetic and trait beta-diversity measures of three animal groups exhibited non-significant linear patterns along the elevational gradient, except for SES . $\beta_{funcsim}$ and SES . $\beta_{funcsor}$ of passerines ($p = 0.003$ and $p < 0.001$) and SES . $\beta_{phylosor}$ of ants ($p = 0.014$) (figure 3 and electronic supplementary material, table S4).

## 4. Discussion

Revealing the general rules of species coexistence and the underlying processes thereof are common goals of community ecologists. With such an ambition, by comparing the

observed patterns (the dissimilarity and the rate of increase in dissimilarity) and the dynamics of niche-based drivers of beta-diversity, we present a stepwise (beta-diversity components, dimensions and species groups) comparative framework for revealing the consistence in the patterns and underlying mechanism of elevational beta-diversity.

Spatial turnover and nestedness are recognized as two antithetic processes leading to community compositional variance along environmental gradients [3]. However, their relative importance often depends on the focal taxon and diversity dimension of interest [21,35,69,70]. Generally, species beta-diversity can be indicative of the species' response to the current environment and/or geographical barrier, whereas trait beta-diversity refers to the taxon-specialized adaptation to the varying environment [35]. The comparative results in three animal groups suggest that species replacement and functional loss/gain may happen similarly in different animal groups responding to the varying environment along an elevational gradient. This is especially true when habitat endemics account for a large proportion of the regional species pool. This contrasting pattern in species and functional beta-diversity has been previously reported in European fish assemblages [34] and South African ant communities [21], implying this pattern is common across ecosystems and taxa. Nevertheless, it is worth noting that the patterns of functional beta-diversity are definitely dependent on the subjective bias (e.g. the selections on trait and diversity metric) and taxon-specialized functional character. Hence, the commonness of the dominion of nestedness in driving functional beta-diversity requires further examinations in various ecosystems.

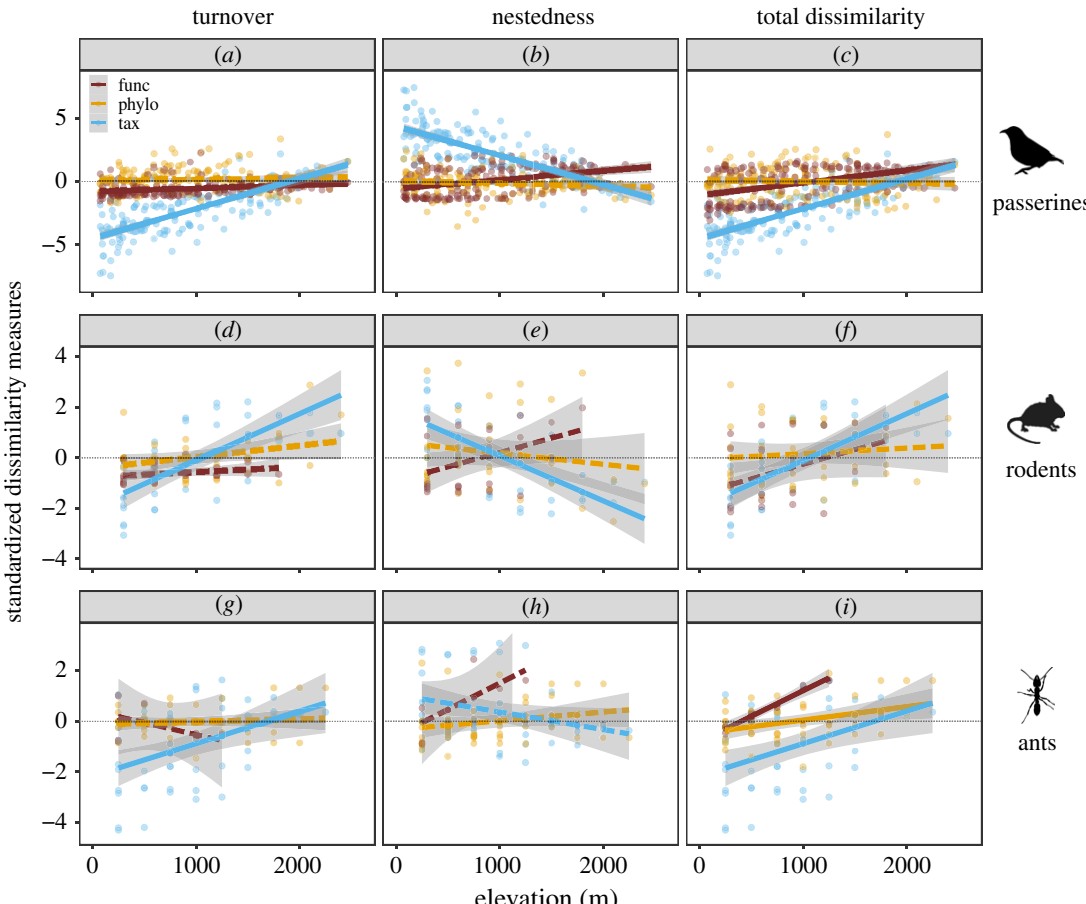

**Figure 3.** Linear elevational patterns of standardized dissimilarities driven by turnover (*a*, *d* and *g*), nestedness (*b*, *e* and *h*) and their combinations (*c*, *f* and *i*) at taxonomic ('tax', light blue (light grey)), phylogenetic ('phylo', orange (grey)) and functional ('func', dark red (dark grey)) dimensions in passerines (*a*–*c*), rodents (*d*–*f*) and ants (*g*–*i*). Solid lines indicate that linear fittings are significant at the level of $p \leq 0.05$, whereas dash lines imply that linear fittings are non-significant ($p > 0.05$). The standardized functional beta-diversity measures of rodents and ants were calculated using subset assemblages containing three or more species. (Online version in colour.)

At the phylogenetic dimension, the comparison between turnover and nestedness-resultant dissimilarity in three animal groups reveals inconsistent patterns, disagreeing with the inference that phylogenetic beta-diversity is dominated by phylogenetic turnover [69,71]. Given that phylogenetic beta-diversity indicates the degree of historical isolation across assemblages at an evolutionary timescale [6,35], the inconsistency at the phylogenetic dimension is understandable. On the one hand, phylogenetic beta-diversity highly depends on the taxon-specialized ecological and evolutionary characteristics (e.g. species pool and phylogenetic scale) [6,7], which explicitly differ across the animal groups involved in this work. On the other hand, the regional evolutionary history has probably left distinct evolutionary imprints on different fauna groups. For instance, the uplift of Qinghai-Tibet Plateau and the glacial-interglacial oscillations have different impacts on homeothermic vertebrates and heterothermic invertebrates via distinct magnitudes of preventing or promoting up-down species exchange [37,72,73]. The accumulation of these complexities might explain the inconsistent patterns of elevational phylogenetic beta-diversity in different animal groups.

The comparisons of beta-diversity across species, phylogenetic and trait dimensions may shed additional light on the processes organizing species [36]. We found that turnover-resultant species dissimilarities in the three species groups are consistently higher than their phylogenetic and functional analogues ($\beta_{sim} > \beta_{phylosim}/\beta_{funcsim}$), whereas functional

nestedness-resultant dissimilarities are consistently higher than their species and phylogenetic analogues ($\beta_{funcsne} > \beta_{phylosne}/\beta_{sne}$). These results imply that the assembling processes have produced different but interdependent outcomes at distinct beta-diversity dimensions. As the major force in structuring elevational beta-diversity at taxonomic dimension, species replacement often occurs between phylogenetic relatives [6,9,36,74] sharing similar functional attributes [34,36], which has produced a high level of functional gain or loss. This evidence reinforces the necessity of integrating multi-dimensional information in estimating assembly processes.

Under the same spatio-temporal extent, higher rates of variation in taxonomic composition suggest higher levels of sensitivity to the changing environment and/or stronger dispersal limitation for the focal species group; higher rates of phylogenetic variations indicate stronger historical isolation, and higher rates of trait variations suggest stronger trait specificity limiting adaptation to different environments [34,35]. At the taxonomic dimension, considering environmental persistence, environmental sensitivity and dispersal ability, the taxonomic dissimilarity of heterothermic invertebrates, such as ants, is as expected to be more sensitive to environmental change than that of homeothermic vertebrates (e.g. passerines and rodents). Unexpectedly, both species turnover and total species dissimilarity in rodents, not ants, displayed higher rates of increase of taxonomic dissimilarity with elevational distance. Occupying different niche positions, passerines, rodents and ants are expected to be

sensitive to the variation of distinct environmental factors along the elevational gradient [9,30] (also supported in our environment dependence analyses). The higher rates of variations in rodent species composition could result from their sensitivity to the biotic and abiotic conditions of their microhabitat (i.e. food resource) [75,76]. Alternatively, the higher rate of increase of species beta-diversity of rodents could be attributed to the effects of species pool and taxonomic scale, which admittedly could affect the patterns of beta-diversity [77–79]. At the phylogenetic dimension, as mentioned before, the distinct evolutionary histories and the inconsistent effect of regional evolutionary events have jointly shaped the taxon-specialized phylogenetic beta-diversity. At increasing elevational distance, the higher rate of increase in the phylogenetic dissimilarity of ants probably results from severer evolutionary isolation and more stable habitat adaptation.

The results of environment dependence analyses reveal the importance of non-random species-environment interactions in affecting species assembly along an elevational gradient. Although the best environmental predictors varied according to the beta-diversity component, diversity dimension and species group, NPP and AMT had the most widespread effect in explaining the elevational patterns of multiple beta-diversity measures. These results indirectly reflect the importance of the primary producer (providing food and habitat) and physiological limitation in organizing animal assemblages. By comparison, owing to the localized monsoon climate and relative high altitude in our study area, AP appears relatively less important for the observed beta-diversity of the three animal groups. Although PET is an index for energy availability similar to temperature sum, it appears less as an interpretation for elevational beta-diversity owing to its low-data quality (extracted from a global data layer with resolution of $1 \, \text{km}^2$). As an air index weakly related to animal life history, it is understandable that AMH displays a weak role in predicting the elevational patterns of animal beta-diversity in this study.

Aiming to assess the dynamics of opposite niche-based processes, we examined the elevational pattern of standardized beta-diversity via linear regressions. Although there was a relatively higher proportion of beta-diversity classified as random, we cannot fully assert that neutral processes are the dominant or only drivers of beta-diversity along this elevational gradient because the linear fitting models of standardized beta-diversity along an elevational gradient show that deterministic processes dynamically transformed with increasing elevational distance, especially for the elevational beta-diversity at taxonomic dimension. The elevational patterns of standardized species turnover reveal that, as elevational distance increased, the force of environmental filtering consistently decreased (e.g. similar environmental filtering across assemblages), while the magnitude of limiting similarity (e.g. negative interspecific competition) gradually increased. Conversely, the decreasing limiting similarity and the increasing environmental filtering jointly contributed to the elevational patterns of nestedness-resultant dissimilarity. It is worth noting that the strength of environmental filtering and negative interspecific effect became balanced in the median elevational distance. In other words, the high proportion of beta-diversity categorized as random could result from the balance between opposite deterministic processes as well as stochasticity. Despite intensive efforts of ecologists,

currently it remains an extraordinary challenge to fully disentangle the effects of opposite deterministic processes (i.e. environmental filtering and interspecific competition) in driving beta-diversity. This methodological flaw of the randomization approach is an urgent issue that needs to be resolved in the future. Nonetheless, this work has represented a substantial improvement to our current knowledge of the general mechanisms underlying multi-dimensional elevational beta-diversity.

## 5. Conclusion

With extensive empirical evidence across three representative animal groups, this work provides a synthetic perspective to understand the underlying processes driving multi-dimensional elevational beta-diversity. We performed a multi-faceted comparative analysis across beta-diversity compositional components (turnover and nestedness), multiple dimensions (species, phylogeny and trait) and species groups (passerines, rodents and ants). The three animal groups were generally consistent on three main points: first, a turnover process dominated the species beta-diversity along an elevational gradient, whereas nestedness process was the main cause for trait-based dissimilarity; second, in the comparison across beta-diversity dimensions, species turnover appeared gradually higher than its phylogenetic and trait-based measures ($\beta_{\text{sim}} > \beta_{\text{phylosim}}/\beta_{\text{funcsim}}$); conversely, functional nestedness was relative higher than its taxonomic and phylogenetic analogues ($\beta_{\text{funcsne}} > \beta_{\text{sne}}/\beta_{\text{phylosne}}$); and third, the relative importance of opposite niche-based processes (environmental filtering and negative competitive exclusion) gradually transformed in driving elevational beta-diversity as the elevational distance increases. Further, via a linear fitting model and forward selection procedure, we found that the elevational beta-diversity patterns and their best environmental predictors both varied across beta-diversity components, dimensions and species groups. Among five environmental factors, NPP and AMT generally performed stronger effects in explaining the elevational patterns of multiple beta-diversity measures. In the comparisons across species groups, rodents were found to display a higher rate of variation in taxonomic dissimilarity with elevational distance, whereas ants exhibited a higher rate of variation in phylogenetic relatedness. Although we have authenticated the dynamics of opposite niche-based processes, currently it is an extraordinary challenge to disentangle the effects of neutral process and the balance of opposing niche-based processes. Despite the complexities and uncertainties in the ecological and evolutionary processes, this study sets a foundation for bettering our understanding of elevational beta-diversity dynamics.

**Ethics.** This study was performed according to the international, national and institutional rules considering animal experiments, clinical studies and biodiversity rights. The study protocol was approved by the Institute of Zoology, Chinese Academy of Sciences (licence: AEI-1203-2013)

**Data accessibility.** All data referred within this study are freely available as follows: sampling information and species composition are available from the Dryad Digital Repository: https://doi.org/10.5061/dryad.mw6m905wf [41]; detailed information of data collection: electronic supplementary material, Appendix S01; phylogeneies: electronic supplementary material, Appendix S02; trait attributes and phylogenetic signal detection: electronic supplementary material, Appendix S03; information of eight auto weather stations: electronic supplementary material, Appendix S04; information for

calculating observed and standardized beta-diversities: electronic supplementary material, Appendix S05; electronic supplementary material, figures and tables: Appendix S06.

Authors' contributions. Y.D., Z.X., Q.Y., F.L. and H.Q. conceived the ideas and designed methodology; Y.D., L.F., Z.X. and Z.W. collected the data; Y.D., T.C. and A.F. analysed the data; Y.D., A.F., J.H. and L.F. led the writing of the manuscript. All authors contributed critically to the drafts and gave final approval for publication.

Competing interests. We declare we have no competing interests

Funding. Y.D. was supported by the Ningxia Hui Autonomous Region Agricultural Science and Technology Independent Innovation Fund (NGSB20211405); L.F. was supported by the Natural Science Foundation of Tibet Autonomous Region of China (XZ2019ZRG-68) and the Biodiversity Investigation, Observation and Assessment Program of Ministry of Ecology and Environment of China; Z.W., A.F., F.L. and Q.Y. were supported by the Second Tibetan Plateau Scientific Expedition and Research Program (STEP; 2019QZKK0501); H.Q. was supported by the National Natural Science Foundation of China (31772432).

Acknowledgements. We appreciate the editor and anonymous reviewers for their constructive comments and suggestions on prior versions of this manuscript. We thank Michael Orr for his kindly help in language polishing and Yan Wu for her suggestions on this manuscript. We also thank the colleagues at SETS (South-East Tibetan plateau Station for integrated observation and research of alpine environment, CAS) and all of other colleagues for their support in fieldwork.

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
