## [Peer Review File · Proceedings of the Royal Society B: Biological Sciences]

Review History

RSPB-2020-2174.R0 (Original submission)

Review form: Reviewer 1

Recommendation

Major revision is needed (please make suggestions in comments)

Scientific importance: Is the manuscript an original and important contribution to its field?

Acceptable

General interest: Is the paper of sufficient general interest?

Good

Quality of the paper: Is the overall quality of the paper suitable?

Acceptable

Is the length of the paper justified?

No

Should the paper be seen by a specialist statistical reviewer?

No

Do you have any concerns about statistical analyses in this paper? If so, please specify them explicitly in your report.

Yes

It is a condition of publication that authors make their supporting data, code and materials available - either as supplementary material or hosted in an external repository. Please rate, if applicable, the supporting data on the following criteria.

Is it accessible?

Yes

Is it clear?

Yes

Is it adequate?

Yes

Do you have any ethical concerns with this paper?

No

Comments to the Author

This manuscript (MS) examined the elevational patterns of beta diversity for three distinctive clades along a large altitudinal gradient in southeast Tibet mountains, China. They decomposed the total beta diversity into components due to species turnover and nestedness for taxonomic, phylogenetic and functional beta diversity, so as to explore the underlying mechanisms for altitudinal diversity patterns. The topic is interesting, and the authors have conducted years of work and have obtained good field data. I also agree with the authors that comparison of altitudinal patterns among different species groups, mountains and diversity components and dimensions is an effective way to explore the underlying mechanisms. However, I have some concerns regarding to the data analyzing method, and the interpretation of the results. Thus the MS still need major revision before it can be accepted for publication.

Major concerns:

1) Data processing. The authors have collected good field-observed data at different altitudes across a large elevational gradients. However, when analyzing altitudinal patterns of beta diversity, they estimated the altitudinal distribution for each species through interpolation. This practice lead to two major weakness: 1) it is well known that these interpolated data will suffered from the problem of the interpolation effect (see papers of e.g. Colwell), because it assumes continuous distribution between the upper and lower limits of a species. Studies have shown that this effect would more or less affect altitudinal richness patterns, but to what extent it would distort beta diversity patterns is still not clear. The MS state that “we applied interpolation correction”, but did not explained how was this done. 2) Using this method, the MS calculated beta diversity for 24-27 elevational bands. However, they actually have observed data for only nine to 17 altitudinal transects. This means that they calculated diversity based on many elevational bands with only estimated species occurrence data.

I would suggest the authors to calculate beta diversity based field data, which is far more reliable than the estimated ones. Actually, your field data is good enough. Though you have fewer altitudinal transects for rodents and ants, the data have covered a similarly large altitudinal gradient as passerines (2000-ca. 4500 m). Since the focus of this MS is to compare the diversity patterns among clades across the whole altitudinal gradient, I think your field data are good enough for this purpose. Anyway, field data are more accurate for a better understanding of the underlying mechanisms.

Meanwhile, the above-mentioned Data processing methods should be included in the Methods section but not in the appendices. Or that the readers may have not noticed how you processed

the data, which is key for the final results.

2) Interpretation of the results. The MS conclude that “it seems challenge to obtain a general beta-diversity rule across distinct components and species groups” (L39). However, it seems at least to me that the results revealed consistent patterns among species groups, and some diversity components, which may suggest some general underlying processes. For instance, most patterns of total and turnover dissimilarity were similar among species groups in Fig. 3, with difference mainly in regression slope (which is natural because you are examine species groups with great differences). Fig. S6 is very interesting, and revealed that both the taxonomic and phylogenetic components of turnover and total dissimilarity increased with longer distance. Meanwhile, these two components also showed consistent humped pattern for nestedness dissimilarity. And these patterns showed good consistence among species groups. The only exception was functional beta diversity, which differed markedly among species groups. But this is not surprising, because you used very different functional traits for different species, which may respond very different to environmental gradients. Thus, it seems at least to me, that your results revealed some general mechanisms underlying altitudinal diversity patterns, though different diversity components and species groups are predicted to show some difference.

The above-mentioned conclusion of the abstract seemed to be mainly based on results such as Tables 1 and Figs. 2 and S5, which compared data of the whole altitudinal gradients together. However, such overall analyses may be misleading. For instance, Fig. S5 showed that random process contributed to large proportion to altitudinal beta diversity patterns, while the effects of two niche-based processes was very weak in many cases. Considering you are examined an altitudinal gradient that is cold enough (2000-4500 m), this result is not easily believed. However, Fig S6 showed that this may be caused by the fact that there is a transition of the dominant role from environmental filtering to interspecific competition with increasing distance in many cases. When the effect of environmental filtering and interspecific competition were similar, the observed beta diversity can be not significantly different from null model expectations, which may not necessarily mean is was a result a neutral processes. Consequently, it may be better for you to report the 95% confidence lines of null models in comparison with observed patterns, which may provide more details of how dominant processes changes with distance or elevation. Thus, results such as Fig. S5 and 3 can provided interesting patterns and explanations, compared with other results. In the present MS, you have focused on other results, which is not interesting enough and made the results and discussions not very informative.

At the same time, it is well known that beta diversity patterns are strongly affected by species numbers. And comparison of observed patterns with null model results is an effective way to distinguish the real beta diversity pattern. However, the present MS is mainly not based on these results, and have only a few discussion on the related results.

3) The introduction is generally well written, especially Fig. 1 (but may need some more explanation on Fig. 1 A and B for readers not familiar with beta diversity to better understand). But the Results and Discussions are not interesting enough and somehow too long (especially some sections of the Discussions), probably because they did not focused on some interesting patterns, such as those mentioned above. The methods to decompose total beta diversity into turnover and nestedness components need to be introduced briefly. Though it has been described in literatures, but this is a key method of the MS and thus need to be introduced for readers to better understand.

Review form: Reviewer 2

Recommendation

Major revision is needed (please make suggestions in comments)

Scientific importance: Is the manuscript an original and important contribution to its field?
Marginal

General interest: Is the paper of sufficient general interest?
Marginal

Quality of the paper: Is the overall quality of the paper suitable?
Poor

Is the length of the paper justified?
Yes

Should the paper be seen by a specialist statistical reviewer?
No

Do you have any concerns about statistical analyses in this paper? If so, please specify them explicitly in your report.
Yes

It is a condition of publication that authors make their supporting data, code and materials available - either as supplementary material or hosted in an external repository. Please rate, if applicable, the supporting data on the following criteria.

Is it accessible?
No

Is it clear?
No

Is it adequate?
No

Do you have any ethical concerns with this paper?
No

Comments to the Author
General

This manuscript by Du et al describes patterns of beta diversity across an elevational gradient in Tibet. The topic is really interesting, and this might be a good dataset to tackle various topics in beta diversity research.

At this stage, however, the manuscript is not ready for publication. It is not clear what the actual goal of the manuscript is, the methods are lacking in important detail, and there is almost no discussion of the ecology of these various animal groups that would help the authors and the reader to understand the patterns that are described.

I have made a range of suggestions below. The biggest task should be to better describe to the reader what has been found before, and what this paper will do to push the field forward. Perhaps the authors already know this, but it does not come across in this version of the manuscript.

Introduction

The introduction touches on many ideas, but doesn't really explain or expand on any of them particularly well. Most importantly, it isn't clear exactly why the authors are interested in the

topic, or what their analyses will add. What gap will the authors fill?

For example, the authors jump around from taxonomic/phylogenetic/functional perspectives on beta diversity, to beta diversity decomposition, to stochastic vs deterministic processes. These are all important topics, but the reader isn't told why they should care. In addition, the authors make many broad statements (with references), but don't unpack what this previous work has found. To my mind, it seems that the central question is really about seeing how consistent all of these different beta diversity patterns (functional vs phylogenetic, or nestedness vs turnover, or stochastic vs deterministic) are across different groups of organisms.

Have other researchers looked at this? What did they find?

With this in mind, maybe it would be worth restructuring the introduction. Some more explanation and description of the wide range of approaches and results would help. Then the reader will understand that the goal in this manuscript is to try and test whether there are any consistent signals in this diversity of results. Can some key example studies be described which find conflicting findings in relation to these beta diversity topics?

Another suggestion: what do the authors predict will happen? Do they expect that functional beta will change much more than phylogenetic beta? Or that some organisms will show less beta than others? If so, why? This will help to add this missing context that I am talking about. These predictions could be discussed toward the end of this section, after the questions are outlined.

Methods

There needs to be more detail about the sampling methods in the main text. Over what timescale were these data collected? Were they the same sites? Were they active or passive sampling methods? Who validated the identifications? We all three taxa sampled across the entire gradient?

Birds: In the supplementary material, the bird sampling section is not detailed enough. What sampling method was used? This is basic information that needs to be in the manuscript.

Rodents: What time scale did this take place over? At one point it seems to be only 2014, but later the text refers to samples from 2018? Were the rodents released or killed? This has important implications for the findings.

Ants: So were the ants actively sampled? Do you think active sampling is a good representation of the relative abundances of the ant community? I am unsure - I think that this method leads to bias as the searcher is drawn to large, colourful or distinctive species. The collection needs to be described in detail in this paper, the authors should not just refer to a citation. In addition, the ants were sampled in a "50*50" but no units are given. Metres?

Elevational distribution interpolation: I imagine that this will have a major influence on any analysis that attempts to understand stochasticity. This procedure deterministically generates a distribution for a species that might not actually be true. It is unclear to me why this is being done in this study, why not just use the observed compositions at each elevation for each animal group? This needs to be discussed.

I do not think that measuring size for each taxon is enough to be called a functional diversity analysis. Functional traits and functional diversity, as written in the introduction, should capture something of the ecological niche of the organism. Using only size, and nothing else, is poor. I would advise reading some of the seminal functional trait papers such as:

McGill, B.J., Enquist, B.J., Weiher, E. and Westoby, M., 2006. Rebuilding community ecology from functional traits. *Trends in ecology & evolution*, 21(4), pp.178-185.

Cadotte, M.W., Carscadden, K. and Mirotnick, N., 2011. Beyond species: functional diversity and the maintenance of ecological processes and services. *Journal of applied ecology*, 48(5),

pp.1079-1087.

Finally, it is not actually stated what constitutes a "community" in this context. Is it the sampling sites? Is it every 100 m of elevation? This is critical information that is needed to assess the manuscript.

Results

The results are presented OK, but I cannot assess how valid they are given the lack of detail in the methods sections. I wondered why there were no plots of the SES analyses?

Discussion

Again, I cannot comment fully on the discussion given that there are major issues with the interpretation of the methods and results.

What is clearly missing, however, is discussion of any ecology. What do these various species and animal groups do? What traits do they have that might drive various beta diversity patterns? Unfortunately, this is probably where the focus on only size related traits will let the manuscript down.

The lack of a clear goal in the introduction also lets this discussion section down. What is the overarching question? The discussion talks a lot about various topics, but it is not clear how they fit together. Do the differences between birds, rodents and ants have anything to say about these patterns? Are these patterns expected or surprising? I suggest that more thought is put into the introduction and the framing of the questions, then this section will be able to flow naturally from that.

Minor

Line 26-28: "The domain process..." I'm not sure what this sentence is saying, can it be rephrased?

Line 28: "Second", but where is the "first" in this abstract?

Line 31-32: "the rates of dissimilarities". I'm not sure what this means. The rate of decay of dissimilarity/similarity with elevational distance? Again, perhaps rephrase.

Line 34: "leading process". I have no idea on this, again, perhaps it is best to make the abstract slightly less technical.

Line 47: "species assembly". I agree, but I think this concept needs to be introduced otherwise it will probably be unclear to the large number of readers of Proceedings B who are not community or macro-ecologists.

Line 56-57: "local and regional processes", sure, but how? This sounds like (and is) an important statement, but it isn't explained properly. I think that concepts like this need to be explained in full if the reader is to understand what the paper is trying to do. Maybe it is unnecessary to mention for the goal at hand?

Line 67: "origin of dissimilarity". I find this a strange phrase. Is it a quote?

Line 95: "rate of dissimilarity". Again, rate relative to what? Change in elevation? Make this clear for the reader.

Line 107: "quantified Lastly". I think there is some text missing here...

Line 121: "hungriness". I don't now what this means.

Line 127: I do not think it is appropriate to put all of the sampling information into the appendix. This needs to be described, at least in brief, in the main text. This is my view, perhaps the editors will disagree.

Line 154: The mean of each measure, or a mean across the 5? This is unclear, but the meanings could be very different.

Line 158: "body size" in ants. How was this measured? There are number of ways to measure body size in ant workers. This detail needs to be in the manuscript, how else could someone else read the paper and replicate the study?!

Line 162: So size is the only functional trait for each species? Do the authors think that size alone captures the ecological niche? What about species that are similar in size but have different diets or habitat preferences that other traits might reveal?

Line 163: Why just the first two components?

Line 171-174: Yes, but you are using a dendrogram method to calculate functional beta diversity, so surely this is not relevant?

Line 177-180: This is all fine, but I think it needs some more explanation for those readers who are not experts in beta diversity decomposition. What exactly does this decomposition do? Is it a standard technique? (I would say that yes it is a standard technique now, but you need to let the reader know this and provide context).

Line 182: I'm not sure if those studies "proposed" that null model. You should find the paper or papers that originally proposed and used it. You should also describe it in more detail in this manuscript - the next few lines do not adequately explain how this procedure "controls for species richness".

Line 205: It isn't really clear to me what "components" and "facets" means in this context.

Line 210-213: Why use regression to test beta diversity against elevational distance, but then a mantel test to test the relationships among beta diversity types? This needs to be explained.

Line 267-269: AICs are not a measure of model fit, and they cannot be used to compare entirely different models sets (i.e. models of turnover vs models of nestedness).

Figure 3: I presume the x-axis is elevational distance? Then why is it log transformed? I don't see why this would be necessary. This is really unclear.

Line 290: Is "analogs" the right word?

Line 322: Do you mean "the main" rather than "domain"?

Line 353-358: Yes, sure. Is that what you found? You need to link this discussion back to your actual results.

Decision letter (RSPB-2020-2174.R0)

23-Oct-2020

Dear Dr Du:

I am writing to inform you that your manuscript RSPB-2020-2174 entitled "A multi-faceted comparative perspective on elevational beta-diversity: the patterns and their causes" has, in its current form, been rejected for publication in Proceedings B.

This action has been taken on the advice of referees, who have recommended that substantial revisions are necessary. With this in mind we would be happy to consider a resubmission, provided the comments of the referees are fully addressed. However please note that this is not a provisional acceptance.

4) Data - please see our policies on data sharing to ensure that you are complying (<https://royalsociety.org/journals/authors/author-guidelines/#data>).

Sincerely,
Dr Sasha Dall
mailto: proceedingsb@royalsociety.org

Associate Editor

Comments to Author:

Using community data of Passerines, rodents and ants sampled on the eastern slope of Mt. Segrila in the southeast Tibetan Plateau, together with phylogenies and functional traits of the sampled species, the authors explored the elevational changes in the taxonomic, phylogenetic and functional beta diversity of the studied groups, and compared the relative effects of deterministic and stochastic processes on the species assembly of these groups. Two reviewers carefully reviewed this manuscript, and both of them thought that this manuscript addressed a very interesting question, and hence could be of general interest for the international readership. I agree with both of them about this. One of the most interesting point is that the authors explored the multidimensional beta diversity of three groups in an integrated framework. Moreover, the community data used in this study are impressive. If these data could be made publically available, this manuscript would make stronger contribution to the field. However, as pointed by both reviewers, the manuscript also suffers from several very serious issues of analyses and writing. Both reviewers provided many very useful comments and suggestions for revision of the manuscript. I urge the authors to consider these comments during their revision. In addition to these comments, I found that the describe of the data collection and processing in the main text is not clear. I suggest to include more details about data collection in the method section. moreover, the authors claimed that they compared the niche-based deterministic processes with stochastic processes. However, none of environmental variables were included in the analyses to demonstrate the relationships between environment and species composition changes.

Reviewer(s)' Comments to Author:

Referee: 1

Comments to the Author(s)

This manuscript (MS) examined the elevational patterns of beta diversity for three distinctive clades along a large altitudinal gradient in southeast Tibet mountains, China. They decomposed the total beta diversity into components due to species turnover and nestedness for taxonomic, phylogenetic and functional beta diversity, so as to explore the underlying mechanisms for altitudinal diversity patterns. The topic is interesting, and the authors have conducted years of work and have obtained good field data. I also agree with the authors that comparison of altitudinal patterns among different species groups, mountains and diversity components and dimensions is an effective way to explore the underlying mechanisms. However, I have some concerns regarding to the data analyzing method, and the interpretation of the results. Thus the MS still need major revision before it can be accepted for publication.

Major concerns:

1) Data processing. The authors have collected good field-observed data at different altitudes across a large elevational gradients. However, when analyzing altitudinal patterns of beta diversity, they estimated the altitudinal distribution for each species through interpolation. This practice lead to two major weakness: 1) it is well known that these interpolated data will suffered

from the problem of the interpolation effect (see papers of e.g. Colwell), because it assumes continuous distribution between the upper and lower limits of a species. Studies have shown that this effect would more or less affect altitudinal richness patterns, but to what extent it would distort beta diversity patterns is still not clear. The MS state that “we applied interpolation correction”, but did not explained how was this done. 2) Using this method, the MS calculated beta diversity for 24-27 elevational bands. However, they actually have observed data for only nine to 17 altitudinal transects. This means that they calculated diversity based on many elevational bands with only estimated species occurrence data.

I would suggest the authors to calculate beta diversity based field data, which is far more reliable than the estimated ones. Actually, your field data is good enough. Though you have fewer altitudinal transects for rodents and ants, the data have covered a similarly large altitudinal gradient as passerines (2000-ca. 4500 m). Since the focus of this MS is to compare the diversity patterns among clades across the whole altitudinal gradient, I think your field data are good enough for this purpose. Anyway, field data are more accurate for a better understanding of the underlying mechanisms.

Meanwhile, the above-mentioned Data processing methods should be included in the Methods section but not in the appendices. Or that the readers may have not noticed how you processed the data, which is key for the final results.

2) Interpretation of the results. The MS conclude that “it seems challenge to obtain a general beta-diversity rule across distinct components and species groups” (L39). However, it seems at least to me that the results revealed consistent patterns among species groups, and some diversity components, which may suggest some general underlying processes. For instance, most patterns of total and turnover dissimilarity were similar among species groups in Fig. 3, with difference mainly in regression slope (which is natural because you are examine species groups with great differences). Fig. S6 is very interesting, and revealed that both the taxonomic and phylogenetic components of turnover and total dissimilarity increased with longer distance. Meanwhile, these two components also showed consistent humped pattern for nestedness dissimilarity. And these patterns showed good consistence among species groups. The only exception was functional beta diversity, which differed markedly among species groups. But this is not surprising, because you used very different functional traits for different species, which may respond very different to environmental gradients. Thus, it seems at least to me, that your results revealed some general mechanisms underlying altitudinal diversity patterns, though different diversity components and species groups are predicted to show some difference.

The above-mentioned conclusion of the abstract seemed to be mainly based on results such as Tables 1 and Figs. 2 and S5, which compared data of the whole altitudinal gradients together. However, such overall analyses may be misleading. For instance, Fig. S5 showed that random process contributed to large proportion to altitudinal beta diversity patterns, while the effects of two niche-based processes was very weak in many cases. Considering you are examined an altitudinal gradient that is cold enough (2000-4500 m), this result is not easily believed. However, Fig S6 showed that this may be caused by the fact that there is a transition of the dominant role from environmental filtering to interspecific competition with increasing distance in many cases. When the effect of environmental filtering and interspecific competition were similar, the observed beta diversity can be not significantly different from null model expectations, which may not necessarily mean is was a result a neutral processes. Consequently, it may be better for you to report the 95% confidence lines of null models in comparison with observed patterns, which may provide more details of how dominant processes changes with distance or elevation. Thus, results such as Fig. S5 and 3 can provided interesting patterns and explanations, compared with other results. In the present MS, you have focused on other results, which is not interesting enough and made the results and discussions not very informative.

At the same time, it is well known that beta diversity patterns are strongly affected by species numbers. And comparison of observed patterns with null model results is an effective way to distinguish the real beta diversity pattern. However, the present MS is mainly not based on these results, and have only a few discussion on the related results.

3) The introduction is generally well written, especially Fig. 1 (but may need some more explanation on Fig. 1 A and B for readers not familiar with beta diversity to better understand). But the Results and Discussions are not interesting enough and somehow too long (especially some sections of the Discussions), probably because they did not focused on some interesting patterns, such as those mentioned above.

The methods to decompose total beta diversity into turnover and nestedness components need to be introduced briefly. Though it has been described in literatures, but this is a key method of the MS and thus need to be introduced for readers to better understand.

Referee: 2

Comments to the Author(s)

General

This manuscript by Du et al describes patterns of beta diversity across an elevational gradient in Tibet. The topic is really interesting, and this might be a good dataset to tackle various topics in beta diversity research.

At this stage, however, the manuscript is not ready for publication. It is not clear what the actual goal of the manuscript is, the methods are lacking in important detail, and there is almost no discussion of the ecology of these various animal groups that would help the authors and the reader to understand the patterns that are described.

I have made a range of suggestions below. The biggest task should be to better describe to the reader what has been found before, and what this paper will do to push the field forward. Perhaps the authors already know this, but it does not come across in this version of the manuscript.

Introduction

The introduction touches on many ideas, but doesn't really explain or expand on any of them particularly well. Most importantly, it isn't clear exactly why the authors are interested in the topic, or what their analyses will add. What gap will the authors fill?

For example, the authors jump around from taxonomic/phylogenetic/functional perspectives on beta diversity, to beta diversity decomposition, to stochastic vs deterministic processes. These are all important topics, but the reader isn't told why they should care. In addition, the authors make many broad statements (with references), but don't unpack what this previous work has found. To my mind, it seems that the central question is really about seeing how consistent all of these different beta diversity patterns (functional vs phylogenetic, or nestedness vs turnover, or stochastic vs deterministic) are across different groups of organisms.

Have other researchers looked at this? What did they find?

With this in mind, maybe it would be worth restructuring the introduction. Some more explanation and description of the wide range of approaches and results would help. Then the reader will understand that the goal in this manuscript is to try and test whether there are any consistent signals in this diversity of results. Can some key example studies be described which find conflicting findings in relation to these beta diversity topics?

Another suggestion: what do the authors predict will happen? Do they expect that functional beta will change much more than phylogenetic beta? Or that some organisms will show less beta than others? If so, why? This will help to add this missing context that I am talking about. These predictions could be discussed toward the end of this section, after the questions are outlined.

Methods

There needs to be more detail about the sampling methods in the main text. Over what timescale were these data collected? Were they the same sites? Were they active or passive sampling methods? Who validated the identifications? We all three taxa sampled across the entire gradient?

Birds: In the supplementary material, the bird sampling section is not detailed enough. What sampling method was used? This is basic information that needs to be in the manuscript.

Rodents: What time scale did this take place over? At one point it seems to be only 2014, but later the text refers to samples from 2018? Were the rodents released or killed? This has important implications for the findings.

Ants: So were the ants actively sampled? Do you think active sampling is a good representation of the relative abundances of the ant community? I am unsure - I think that this method leads to bias as the searcher is drawn to large, colourful or distinctive species. The collection needs to be described in detail in this paper, the authors should not just refer to a citation. In addition, the ants were sampled in a "50*50" but no units are given. Metres?

Elevational distribution interpolation: I imagine that this will have a major influence on any analysis that attempts to understand stochasticity. This procedure deterministically generates a distribution for a species that might not actually be true. It is unclear to me why this is being done in this study, why not just use the observed compositions at each elevation for each animal group? This needs to be discussed.

I do not think that measuring size for each taxon is enough to be called a functional diversity analysis. Functional traits and functional diversity, as written in the introduction, should capture something of the ecological niche of the organism. Using only size, and nothing else, is poor. I would advise reading some of the seminal functional trait papers such as:

McGill, B.J., Enquist, B.J., Weiher, E. and Westoby, M., 2006. Rebuilding community ecology from functional traits. *Trends in ecology & evolution*, 21(4), pp.178-185.

Cadotte, M.W., Carscadden, K. and Mirotnick, N., 2011. Beyond species: functional diversity and the maintenance of ecological processes and services. *Journal of applied ecology*, 48(5), pp.1079-1087.

Finally, it is not actually stated what constitutes a "community" in this context. Is it the sampling sites? Is it every 100 m of elevation? This is critical information that is needed to assess the manuscript.

Results

The results are presented OK, but I cannot assess how valid they are given the lack of detail in the methods sections. I wondered why there were no plots of the SES analyses?

Discussion

Again, I cannot comment fully on the discussion given that there are major issues with the interpretation of the methods and results.

What is clearly missing, however, is discussion of any ecology. What do these various species and animal groups do? What traits do they have that might drive various beta diversity patterns? Unfortunately, this is probably where the focus on only size related traits will let the manuscript down.

The lack of a clear goal in the introduction also lets this discussion section down. What is the overarching question? The discussion talks a lot about various topics, but it is not clear how they fit together. Do the differences between birds, rodents and ants have anything to say about these patterns? Are these patterns expected or surprising? I suggest that more thought is put into the

introduction and the framing of the questions, then this section will be able to flow naturally from that.

Minor

Line 26-28: "The domain process..." I'm not sure what this sentence is saying, can it be rephrased?

Line 28: "Second", but where is the "first" in this abstract?

Line 31-32: "the rates of dissimilarities". I'm not sure what this means. The rate of decay of dissimilarity/similarity with elevational distance? Again, perhaps rephrase.

Line 34: "leading process". I have no idea on this, again, perhaps it is best to make the abstract slightly less technical.

Line 47: "species assembly". I agree, but I think this concept needs to be introduced otherwise it will probably be unclear to the large number of readers of Proceedings B who are not community or macro-ecologists.

Line 56-57: "local and regional processes", sure, but how? This sounds like (and is) an important statement, but it isn't explained properly. I think that concepts like this need to be explained in full if the reader is to understand what the paper is trying to do. Maybe it is unnecessary to mention for the goal at hand?

Line 67: "origin of dissimilarity". I find this a strange phrase. Is it a quote?

Line 95: "rate of dissimilarity". Again, rate relative to what? Change in elevation? Make this clear for the reader.

Line 107: "quantified Lastly". I think there is some text missing here...

Line 121: "hungriness". I don't know what this means.

Line 127: I do not think it is appropriate to put all of the sampling information into the appendix. This needs to be described, at least in brief, in the main text. This is my view, perhaps the editors will disagree.

Line 154: The mean of each measure, or a mean across the 5? This is unclear, but the meanings could be very different.

Line 158: "body size" in ants. How was this measured? There are number of ways to measure body size in ant workers. This detail needs to be in the manuscript, how else could someone else read the paper and replicate the study?!

Line 162: So size is the only functional trait for each species? Do the authors think that size alone captures the ecological niche? What about species that are similar in size but have different diets or habitat preferences that other traits might reveal?

Line 163: Why just the first two components?

Line 171-174: Yes, but you are using a dendrogram method to calculate functional beta diversity, so surely this is not relevant?

Line 177-180: This is all fine, but I think it needs some more explanation for those readers who are not experts in beta diversity decomposition. What exactly does this decomposition do? Is it a standard technique? (I would say that yes it is a standard technique now, but you need to let the reader know this and provide context).

Line 182: I'm not sure if those studies "proposed" that null model. You should find the paper or papers that originally proposed and used it. You should also describe it in more detail in this manuscript - the next few lines do not adequately explain how this procedure "controls for species richness".

Line 205: It isn't really clear to me what "components" and "facets" means in this context.

Line 210-213: Why use regression to test beta diversity against elevational distance, but then a mantel test to test the relationships among beta diversity types? This needs to be explained.

Line 267-269: AICs are not a measure of model fit, and they cannot be used to compare entirely different models sets (i.e. models of turnover vs models of nestedness).

Figure 3: I presume the x-axis is elevational distance? Then why is it log transformed? I don't see why this would be necessary. This is really unclear.

Line 290: Is "analogs" the right word?

Line 322: Do you mean "the main" rather than "domain"?

Line 353-358: Yes, sure. Is that what you found? You need to link this discussion back to your actual results.

Author's Response to Decision Letter for (RSPB-2020-2174.R0)

See Appendix A.

RSPB-2020-3066.R0

Review form: Reviewer 1

Recommendation

Major revision is needed (please make suggestions in comments)

Scientific importance: Is the manuscript an original and important contribution to its field?

Good

General interest: Is the paper of sufficient general interest?

Good

Quality of the paper: Is the overall quality of the paper suitable?

Acceptable

Is the length of the paper justified?

Yes

Should the paper be seen by a specialist statistical reviewer?

No

Do you have any concerns about statistical analyses in this paper? If so, please specify them explicitly in your report.

No

It is a condition of publication that authors make their supporting data, code and materials available - either as supplementary material or hosted in an external repository. Please rate, if applicable, the supporting data on the following criteria.

Is it accessible?

Yes

Is it clear?

Yes

Is it adequate?

Yes

Do you have any ethical concerns with this paper?

No

Comments to the Author

Comments to the Author(s)

This manuscript (MS) has been substantially revised, especially in the way to analyze data and to interpret the results. I am glad to see the authors payed more attention to summarize general patterns from their results, which is an important aim for us to compare the patterns from different species groups and diversity components and dimensions. While the MS has been

clearly improved, there are still some points that needed to be clarified, or improved, as summarized below.

Major concerns:

1) L33: In abstract, you stated that: “deterministic and neutral processes have jointly contributed to driving community beta-diversity, ...”. However, as mentioned last time, the large proportion of contribution of the so-called random processes in Fig. S4 may simply be a result of transition from environmental filtering to negative interspecific effect, which causes a large proportion around the middle elevational distance not distinguishable from null model predictions (this can be somehow indicated from the new Fig. 3, though Fig. 3 now included too many lines and some patterns were not clear. I would suggested you to split Fig. 3 in to several ones and move Fig. 3 into appendices, because it seems that you did not mentioned Fig. 3 in the new MS and the related results were based on Fig. S5). It is nice that you have discussed about this point at the end of Discussion section. Thus it is not appropriate for you to conclude the above-mentioned sentence in the Abstract. Also, in the middle panel of Fig. 1F (and the related words in texts), this point should also be clarified (e.g. L247 is nice, but L117 and L80-94 may lead some readers to interpret your Fig. S4 as largely caused by neutral processes). I believe this is important, because your results till now can not prove that neutral processes played a so important role as suggested in Fig. S4. For instance, in most part of the Discussions you explained your elevational patterns with deterministic processes (e.g. L356-363 are well written). These explanations based on deterministic processes are in contradiction with the high contribution of neutral processes as suggested in Fig. S4. So I believe it is better for you to revise related word throughout the MS, or, I am not sure if you really need Fig. S4.

2) The MS has provided many results in the appendices, but it seems that not all of them were necessary, which made the MS complicated. It also seems that some of the results were not mentioned in the Discussions at all. Please consider to remain only the most important ones.

Meanwhile, the connections between your results and discussions are not clear. For instance, L384 said that “Our analysis on the best environmental predictors supports this”. However, how it was supported was not clearly explained, and it is easily to made the readers confused. Frankly, I did not find interesting discussions about the results in Table S3. It’s good and important to relate diversity patterns to climate factors, in addition to elevational gradients. But without good discussions and interpretations, Table S3 seems not necessary. For instance, why NPP and temperature is more important in explaining beta diversity patterns than precipitation and humidity? This is an interesting results, but was not well explained.

Generally speaking, the Introduction and methods sections are well written now, but the Results and Discussion sections are not well organized and sometimes hard to read. Please revise so that the MS can be clear in expressing your logics.

Minor points:

L25: remain poorly understood. This is clearly not appropriate.

L30: has dominated species beta-diversity -> has dominated altitudinal patterns of species beta-diversity in your study.

L56: cannot account for ecological and evolutionary differentiation-> cannot account for functional and evolutionary differentiation

Review form: Reviewer 3

Recommendation

Major revision is needed (please make suggestions in comments)

Scientific importance: Is the manuscript an original and important contribution to its field?
Good

General interest: Is the paper of sufficient general interest?
Excellent

Quality of the paper: Is the overall quality of the paper suitable?
Good

Is the length of the paper justified?
Yes

Should the paper be seen by a specialist statistical reviewer?
No

Do you have any concerns about statistical analyses in this paper? If so, please specify them explicitly in your report.
No

It is a condition of publication that authors make their supporting data, code and materials available - either as supplementary material or hosted in an external repository. Please rate, if applicable, the supporting data on the following criteria.

Is it accessible?
Yes

Is it clear?
Yes

Is it adequate?
Yes

Do you have any ethical concerns with this paper?
No

Comments to the Author

As a new referee, I read carefully this resubmitted MS. While the authors has done a very good revision work for several issues, i have major concerns about the use of functional dendrogram to calculate functional beta diversity.

I am concern that the authors did not provide a quantitative assesment of how functional distances are faithfully represented in the functional dendrogram, while recent studies showed that this represents an important step with strong consequences when quantifying functional diversity (see Maire et al. 2015). The authors should show the relationship between the distances derived from the functional dendrogram and the original Euclidean distance matrix (one might expect a strong triangular relationship with closely related species being far apart in the functional dendrogram) and (ii) used a metric, i.e. not the correlation coefficient that is biased but the Norm2 of Merigot et al. (2010) in Ecology or mSD of Maire et al. (2015) in Global Ecology and Biogeography

The main point is that you do not only loss information using a functional dendrogram but most importantly, you provide spurious functional distances between species, with a strong triangular relationship between the cophenetic distances derived from the functional dendrogram and the original euclidean distances (see Fig 3 in Maire et al. 2015). In contrast, using a functional space, you only loss information but this loss is limited with the increasing number of axes used to build the functional space (see again Fig 3 with the mean squared deviation, mSD decreasing with

increasing number of axes). That's why it is an important step to determine the number of axes to be used to derive the FD metrics. It is a « compromise » between the quality of the representation of the original distances and the time to compute the indices, but see Maire et al. 2015 to see how to define it.

I agree that a functional space-based approach has limitations such as samples with lower number of species than the number of selected PCA or PCOA axes cannot be used ! But, at least, the authors could have calculated functional beta diversity metrics based on a functional space approach using a subset of data (i.e. only considering samples with 3 or more species), and then calculate the correlation (Mantel test using Pearson or Spearman's metrics) between the two matrices, in order to show whether the two approaches lead to similar conclusions (See Villeger et al. 2013 that showed that these two approaches can lead to contrasting conclusions or Villeger et al. 2018 in *Ecology Letters*). In addition, the justifications made by the authors are only included in the supplementary materials, with no reference in the main text (in the methodological part).

Overall, I have strong doubt about the results obtained using a dendrogram-based approach because the authors did not demonstrate, quantitatively, that the beta-diversity metrics based on a functional space and a dendrogram, respectively, lead to similar results ! If the authors can demonstrate that their functional dendrogram based approach did not affect the results and conclusions, I think that this study can represent a significant contribution to the field of large-scale community ecology.

References :

Villéger S., Maire E., Leprieur F. 2017. On the risks of using dendrograms to measure functional diversity and multidimensional spaces to measure phylogenetic diversity: a comment on Sobral et al. (2016). *Ecology Letters*. 20: 554–557.

Maire E., Grenouillet G., Brosse S. & Villéger S. 2015. How many dimensions to accurately assess functional diversity? A pragmatic approach for assessing the quality of functional spaces. *Global Ecology and Biogeography*. 24: 728–740.

Villéger S., Grenouillet G. & Brosse S. 2013. Decomposing functional β -diversity reveals that low functional β -diversity is driven by low functional turnover in European fish assemblages. *Global Ecology and Biogeography*. 22: 671–681.

Decision letter (RSPB-2020-3066.R0)

29-Jan-2021

I am writing to inform you that this version of your manuscript RSPB-2020-3066 entitled "A multi-faceted comparative perspective on elevational beta-diversity: the patterns and their causes" has, in its current form, been rejected for publication in *Proceedings B*.

This action has been taken on the advice of referees, who have recommended that substantial revisions are necessary. With this in mind we would be happy to consider a resubmission, provided the comments of the referees are fully addressed. However please note that this is not a provisional acceptance.

The resubmission will be treated as a new manuscript. However, we will approach the same reviewers if they are available and it is deemed appropriate to do so by the Editor. Please note that resubmissions must be submitted within six months of the date of this email. In exceptional

circumstances, extensions may be possible if agreed with the Editorial Office. Manuscripts submitted after this date will be automatically rejected.

- 1) A 'response to referees' document including details of how you have responded to the comments, and the adjustments you have made.
- 2) A clean copy of the manuscript and one with 'tracked changes' indicating your 'response to referees' comments document.
- 3) Line numbers in your main document.
- 4) Please read our data sharing policies to ensure that you meet our requirements <https://royalsociety.org/journals/authors/author-guidelines/#data>.

Sincerely,
Dr Sasha Dall
mailto:proceedingsb@royalsociety.org

Reviewer(s)' Comments to Author:

Referee: 1

Comments to the Author(s).

Comments to the Author(s)

This manuscript (MS) has been substantially revised, especially in the way to analyze data and to interpret the results. I am glad to see the authors payed more attention to summarize general patterns from their results, which is an important aim for us to compare the patterns from different species groups and diversity components and dimensions. While the MS has been clearly improved, there are still some points that needed to be clarified, or improved, as summarized below.

Major concerns:

1) L33: In abstract, you stated that: "deterministic and neutral processes have jointly contributed to driving community beta-diversity, ...". However, as mentioned last time, the large proportion of contribution of the so-called random processes in Fig. S4 may simply be a result of transition from environmental filtering to negative interspecific effect, which causes a large proportion around the middle elevational distance not distinguishable from null model predictions (this can be somehow indicated from the new Fig. 3, though Fig. 3 now included too many lines and some patterns were not clear. I would suggested you to split Fig. 3 in to several ones and move Fig. 3 into appendices, because it seems that you did not mentioned Fig. 3 in the new MS and the related results were based on Fig. S5). It is nice that you have discussed about this point at the end of Discussion section. Thus it is not appropriate for you to conclude the above-mentioned sentence in the Abstract. Also, in the middle panel of Fig. 1F (and the related words in texts), this point should also be clarified (e.g. L247 is nice, but L117 and L80-94 may lead some readers to interpret your Fig. S4 as largely caused by neutral processes). I believe this is important, because your results till now can not prove that neutral processes played a so important role as suggested in Fig. S4. For instance, in most part of the Discussions you explained your elevational patterns with deterministic processes (e.g. L356-363 are well written). These explanations based on deterministic processes are in contradiction with the high contribution of neutral processes as suggested in Fig. S4. So I believe it is better for you to revise related word throughout the MS, or, I am not sure if you really need Fig. S4.

2) The MS has provided many results in the appendices, but it seems that not all of them were necessary, which made the MS complicated. It also seems that some of the results were not mentioned in the Discussions at all. Please consider to remain only the most important ones.

Meanwhile, the connections between your results and discussions are not clear. For instance, L384 said that "Our analysis on the best environmental predictors supports this". However, how it was supported was not clearly explained, and it is easily to made the readers confused.

Frankly, I did not find interesting discussions about the results in Table S3. It's good and important to relate diversity patterns to climate factors, in addition to elevational gradients. But without good discussions and interpretations, Table S3 seems not necessary. For instance, why NPP and temperature is more important in explaining beta diversity patterns than precipitation and humidity? This is an interesting results, but was not well explained.

Generally speaking, the Introduction and methods sections are well written now, but the Results and Discussion sections are not well organized and sometimes hard to read. Please revise so that the MS can be clear in expressing your logics.

Minor points:

L25: remain poorly understood. This is clearly not appropriate.

L30: has dominated species beta-diversity -> has dominated altitudinal patterns of species beta-diversity in your study.

L56: cannot account for ecological and evolutionary differentiation-> cannot account for functional and evolutionary differentiation

Referee: 3

Comments to the Author(s).

As a new referee, I read carefully this resubmitted MS. While the authors has done a very good revision work for several issues, i have major concerns about the use of functional dendrogram to calculate functional beta diversity.

I am concern that the authors did not provide a quantitative assesment of how functional distances are faithfully represented in the functional dendrogram, while recent studies showed that this represents an important step with strong consequences when quantifying functional diversity (see Maire et al. 2015). The authors should show the relationship between the distances derived from the functional dendrogram and the original Euclidean distance matrix (one might expect a strong triangular relationship with closely related species being far apart in the functional dendrogram) and (ii) used a metric, i.e. not the correlation coefficient that is biased but the Norm2 of Merigot et al. (2010) in Ecology or mSD of Maire et al. (2015) in Global Ecology and Biogeography

The main point is that you do not only loss information using a functional dendrogram but most importantly, you provide spurious functional distances between species, with a strong triangular relationship between the cophenetic distances derived from the functional dendrogram and the original euclidean distances (see Fig 3 in Maire et al. 2015). In contrast, using a functional space, you only loss information but this loss is limited with the increasing number of axes used to build the functional space (see again Fig 3 with the mean squared deviation, mSD decreasing with increasing number of axes). That's why it is an important step to determine the number of axes to be used to derived the FD metrics. It is a « compromise » between the quality of the representation of the original distances and the time to compute the indices, but see Maire et al. 2015 to see how to define it.

I agree that a functional space-based approach has limitations such as samples with lower number of species than the number of selected PCA or PCOA axes cannot be used ! But, at least, the authors could have calculated functional beta diversity metrics based on a functional space approach using a subset of data (i.e. only considering samples with 3 or more species), and then calculate the correlation (Mantel test using Pearson or Spearman's metrics) between the two

matrices, in order to show whether the two approaches lead to similar conclusions (See Villeger et al. 2013 that showed that these two approaches can lead to contrasting conclusions or Villeger et al. 2018 in *Ecology Letters*). In addition, the justifications made by the authors are only included in the supplementary materials, with no reference in the main text (in the methodological part).

Overall, i have strong doubt about the results obtained using a dendrogram-based approach because the authors did not demonstrate, quantitatively, that the beta-diversity metrics based on a functional space and a dendrogram, respectively, lead to similar results ! If the authors can demonstrate that their functional dendrogram based approach did not affect the results and conclusions, i think that this study can represent a significant contribution to the field of large-scale community ecology.

References :

Villéger S., Maire E., Leprieur F. 2017. On the risks of using dendrograms to measure functional diversity and multidimensional spaces to measure phylogenetic diversity: a comment on Sobral et al. (2016). *Ecology Letters*. 20: 554–557.

Maire E., Grenouillet G., Brosse S. & Villéger S. 2015. How many dimensions to accurately assess functional diversity? A pragmatic approach for assessing the quality of functional spaces. *Global Ecology and Biogeography*. 24: 728–740.

Villéger S., Grenouillet G. & Brosse S. 2013. Decomposing functional β -diversity reveals that low functional β -diversity is driven by low functional turnover in European fish assemblages. *Global Ecology and Biogeography*. 22: 671–681.

Author's Response to Decision Letter for (RSPB-2020-3066.R0)

See Appendix B.

RSPB-2021-0343.R0

Review form: Reviewer 1

Recommendation

Accept with minor revision (please list in comments)

Scientific importance: Is the manuscript an original and important contribution to its field?

Acceptable

General interest: Is the paper of sufficient general interest?

Good

Quality of the paper: Is the overall quality of the paper suitable?

Good

Is the length of the paper justified?

Yes

Should the paper be seen by a specialist statistical reviewer?

No

Do you have any concerns about statistical analyses in this paper? If so, please specify them explicitly in your report.

No

It is a condition of publication that authors make their supporting data, code and materials available - either as supplementary material or hosted in an external repository. Please rate, if applicable, the supporting data on the following criteria.

Is it accessible?

Yes

Is it clear?

Yes

Is it adequate?

Yes

Do you have any ethical concerns with this paper?

No

Comments to the Author

This manuscript (MS) has been improved again. I do not have further major concerns except for a few minor points listed below.

L37~38: Maybe it is better to said something like: the effect of A increased with as elevational distance increased, while that of B decrease, which lead to ..., so that the readers can get more clear information.

A related point is the change to the use of "limiting/promoting dissimilarity" in the new MS (e.g. Fig. 1 and many sentences elsewhere). These terms are less easily to be understood than the commonly used "environmental filtering vs. limiting similarity", especially because limiting dissimilarity has converse meaning with the widely used "limiting similarity". This markedly decreased the readability of some sentences (e.g. "Conversely, the decreasing promoting dissimilarity and the increasing limiting dissimilarity jointly contributed to the elevational patterns of nestedness-resultant dissimilarity". This sentence in Discussions is hard to understand). If there's no essential difference, I would suggest to use the commonly used terms. However, this is only a suggestion.

L399: "By comparison, due to the localized monsoon climate and relative high altitude in our study area, AP and PET appear relatively less important for the observed beta-diversity of three animal groups."

PET is an index for energy availability, and is similar as temperature sum. This sentence may give some readers an impression that PET is similar as AP. In you data, PET and NPP were extracted from global dataset, while AMT, AP and AMH were based on field-measured data. So it is possible that the low explanatory power of PET is because you PET data is not accurate enough compared with AMT.

Review form: Reviewer 3

Recommendation

Accept as is

Scientific importance: Is the manuscript an original and important contribution to its field?

Excellent

General interest: Is the paper of sufficient general interest?

Good

Quality of the paper: Is the overall quality of the paper suitable?

Good

Is the length of the paper justified?

Yes

Should the paper be seen by a specialist statistical reviewer?

No

Do you have any concerns about statistical analyses in this paper? If so, please specify them explicitly in your report.

No

It is a condition of publication that authors make their supporting data, code and materials available - either as supplementary material or hosted in an external repository. Please rate, if applicable, the supporting data on the following criteria.

Is it accessible?

Yes

Is it clear?

Yes

Is it adequate?

Yes

Do you have any ethical concerns with this paper?

No

Comments to the Author

no further comments

Decision letter (RSPB-2021-0343.R0)

19-Mar-2021

Dear Dr Du

I am pleased to inform you that your manuscript RSPB-2021-0343 entitled "A multi-faceted comparative perspective on elevational beta-diversity: the patterns and their causes" has been accepted for publication in Proceedings B.

The referee(s) have recommended publication, but also suggest some minor revisions to your manuscript. Therefore, I invite you to respond to the referee(s)' comments and revise your manuscript. Because the schedule for publication is very tight, it is a condition of publication that you submit the revised version of your manuscript within 7 days. If you do not think you will be able to meet this date please let us know.

Sincerely,
Dr Sasha Dall
mailto:proceedingsb@royalsociety.org

Reviewer(s)' Comments to Author:

Referee: 3

Comments to the Author(s).

no further comments

Referee: 1

Comments to the Author(s).

This manuscript (MS) has been improved again. I do not have further major concerns except for a few minor points listed below.

L37~38: Maybe it is better to said something like: the effect of A increased with as elevational distance increased, while that of B decrease, which lead to ..., so that the readers can get more clear information.

A related point is the change to the use of “limiting/promoting dissimilarity” in the new MS (e.g. Fig. 1 and many sentences elsewhere). These terms are less easily to be understood than the commonly used “environmental filtering vs. limiting similarity”, especially because limiting dissimilarity has converse meaning with the widely used “limiting similarity”. This markedly decreased the readability of some sentences (e.g. “Conversely, the decreasing promoting dissimilarity and the increasing limiting dissimilarity jointly contributed to the elevational patterns of nestedness-resultant dissimilarity”. This sentence in Discussions is hard to understand). If there’s no essential difference, I would suggest to use the commonly used terms. However, this is only a suggestion.

L399: “By comparison, due to the localized monsoon climate and relative high altitude in our study area, AP and PET appear relatively less important for the observed beta-diversity of three animal groups.”

PET is an index for energy availability, and is similar as temperature sum. This sentence may give some readers an impression that PET is similar as AP. In you data, PET and NPP were extracted

from global dataset, while AMT, AP and AMH were based on field-measured data. So it is possible that the low explanatory power of PET is because you PET data is not accurate enough compared with AMT.

Author's Response to Decision Letter for (RSPB-2021-0343.R0)

See Appendix C.

Decision letter (RSPB-2021-0343.R1)

22-Mar-2021

Dear Dr Du

I am pleased to inform you that your manuscript entitled "A multi-faceted comparative perspective on elevational beta-diversity: the patterns and their causes" has been accepted for publication in Proceedings B.

Data Accessibility section

Open Access

Paper charges

All supplementary materials accompanying an accepted article will be treated as in their final form. They will be published alongside the paper on the journal website and posted on the online

figshare repository. Files on figshare will be made available approximately one week before the accompanying article so that the supplementary material can be attributed a unique DOI.

Sincerely,
Proceedings B
<mailto:proceedingsb@royalsociety.org>

Appendix A

Responding to Editor's and Reviewer's comments

Associate Editor:

Editor's comments: Using community data of Passerines, rodents and ants sampled on the eastern slope of Mt. Segrila in the southeast Tibetan Plateau, together with phylogenies and functional traits of the sampled species, the authors explored the elevational changes in the taxonomic, phylogenetic and functional beta diversity of the studied groups, and compared the relative effects of deterministic and stochastic processes on the species assembly of these groups. Two reviewers carefully reviewed this manuscript, and both of them thought that this manuscript addressed a very interesting question, and hence could be of general interest for the international readership. I agree with both of them about this. One of the most interesting point is that the authors explored the multidimensional beta diversity of three groups in an integrated framework. Moreover, the community data used in this study are impressive. If these data could be made publically available, this manuscript would make stronger contribution to the field.

***Response:** We deeply appreciate for Editor's constructive comments and suggestions for this manuscript. According to the Editor's comments, we have revised this manuscript carefully. For the data sharing and open access, all the authors totally agree with it. The full raw dataset of community composition of three animal groups has been uploaded to Dryad ([doi:10.5061/dryad.k98sf7m50](https://doi.org/10.5061/dryad.k98sf7m50)) (Lines 456-464).*

Editor's comments: However, as pointed by both reviewers, the manuscript also suffers from several very serious issues of analyses and writing. Both reviewers provided many very useful comments and suggestions for revision of the manuscript. I urge the authors to consider these comments during their revision.

***Response:** According to Editor and two reviewers' comments and suggestions, the manuscript has been revised line-by-line. To solve the grammatical problems, we invited a native speaker, Dr. Michael Orr, to revise it. All of the traces of revision and*

correction were in the “tracked changes” version of the manuscript.

Editor’s comments: In addition to these comments, I found that the description of the data collection and processing in the main text is not clear. I suggest to include more details about data collection in the method section.

Response: Due to the word limit, we didn’t clarify the information of data collection and processing in the main text in the former version. In the new one, we provided a concise description of data collection in the M&M section (Lines 133-155). The detailed description of data collection was in the Appendix file 01.

Editor’s comments: Moreover, the authors claimed that they compared the niche-based deterministic processes with stochastic processes. However, none of environmental variables were included in the analyses to demonstrate the relationships between environment and species composition changes.

Response: Thanks for the useful suggestion. We added relevant analyses to examine the relationships between local environment and elevational beta-diversity. We extracted five environmental variables including three climate factors (annual mean temperature, AMT; annual precipitation, AP; and annual mean humidity, AMH) from climatic records of local auto weather stations, and two MODIS variables (net primary production, NPP and potential evapotranspiration, PET). Based on a forward selection procedure of linear regression models, the best environmental predictors have been selected for each beta-diversity component in three animal groups. Detailed information of all the analysis above was in the parts of “Environmental variables” and “Statistical analyses” in the section of “M&M” (Lines 203-216, 235-237). The detailed information of local auto weather stations was in Appendix file 03.

Reviewer #1:

Reviewer’s comments: This manuscript (MS) examined the elevational patterns of beta diversity for three distinctive clades along a large altitudinal gradient in southeast

Tibet mountains, China. They decomposed the total beta diversity into components due to species turnover and nestedness for taxonomic, phylogenetic and functional beta diversity, so as to explore the underlying mechanisms for altitudinal diversity patterns. The topic is interesting, and the authors have conducted years of work and have obtained good field data. I also agree with the authors that comparison of altitudinal patterns among different species groups, mountains and diversity components and dimensions is an effective way to explore the underlying mechanisms. However, I have some concerns regarding to the data analyzing method, and the interpretation of the results. Thus the MS still need major revision before it can be accepted for publication.

Response: *Thanks for all the constructive comments and suggestions. We revised the manuscript carefully, and response to the comments point by point in the following paragraphs. We also uploaded a “tracked changes” version of the manuscript to make reviewing easier.*

Reviewer’s comments: Major concerns: 1) Data processing. The authors have collected good field-observed data at different altitudes across a large elevational gradients. However, when analyzing altitudinal patterns of beta diversity, they estimated the altitudinal distribution for each species through interpolation. This practice lead to two major weakness: 1) it is well known that these interpolated data will suffered from the problem of the interpolation effect (see papers of e.g. Colwell), because it assumes continuous distribution between the upper and lower limits of a species. Studies have shown that this effect would more or less affect altitudinal richness patterns, but to what extent it would distort beta diversity patterns is still not clear. The MS state that “we applied interpolation correction”, but did not explained how was this done. 2) Using this method, the MS calculated beta diversity for 24-27 elevational bands. However, they actually have observed data for only nine to 17 altitudinal transects. This means that they calculated diversity based on many elevational bands with only estimated species occurrence data.

Response: *Thank a lot for pointing out the problem about data collection and*

processing. In the previous version of this manuscript, we applied a linear interpolation for each species along elevational gradient for the beta-diversity analyses, which was not a suitable way to handle the data.

Herein, we agree with the two major weaknesses pointed out by the reviewer. We reran all of the analyses based on the raw presence-absence community data (Lines 219-221). Based on raw community data, we have detected three general points in the patterns and underlying mechanism of elevational beta-diversity: first, turnover process dominated elevational beta-diversity at taxonomic dimension; second, the taxonomic turnover and total dissimilarity were successively higher than their phylogenetic and functional analogues; third, deterministic and neutral processes have jointly contributed to driving community beta-diversity, with the dominant process gradually changing as elevational distance increased (Lines 29-35). More details on the beta-diversity calculation and decomposition were in the appendices (Appendix file 05).

Reviewer's comments: I would suggest the authors to calculate beta diversity based field data, which is far more reliable than the estimated ones. Actually, your field data is good enough. Though you have fewer altitudinal transects for rodents and ants, the data have covered a similarly large altitudinal gradient as passerines (2000-ca. 4500 m). Since the focus of this MS is to compare the diversity patterns among clades across the whole altitudinal gradient, I think your field data are good enough for this purpose. Anyway, field data are more accurate for a better understanding of the underlying mechanisms.

Response: *We totally agree with your comment, “**field data are more accurate for a better understanding of the underlying mechanisms**”. Moreover, the interpolation along elevational gradient possibly reduces the dissimilarity across assemblages via increasing the proportion of common species. We recalculated the beta diversity based on field data directly. Due to the changes in community data, the results of observed and standardized beta-diversity, as well as the comparative analyses, have been revised. All the details were in the “tracked changes” version of the manuscript.*

Reviewer's comments: Meanwhile, the above-mentioned Data processing methods should be included in the Methods section but not in the appendices. Or that the readers may have not noticed how you processed the data, which is key for the final results.

Response: Considering the word limit, we put detailed information on data processing in the appendices. Following your suggestion, we provided a concise information about data collections in M&M (Lines 133-155). Since we used the raw absence-presence community data in further beta-analyses, the original "data processing" in appendix file was out of fashion now. We removed the entire related text in this version. Besides, the part of "Observed and standardized measurements of pairwise beta-diversity" was updated according to reviewer's comments. Due to word limit in PRSB, a concise description of calculating observed and standardized beta-diversity was in the part of "Statistical analyses" (Lines 219-226). A more detailed information of this part was loaded in appendices (Appendix file 05).

Reviewer's comments: 2) Interpretation of the results. The MS conclude that "it seems challenge to obtain a general beta-diversity rule across distinct components and species groups" (L39). However, it seems at least to me that the results revealed consistent patterns among species groups, and some diversity components, which may suggest some general underlying processes. For instance, most patterns of total and turnover dissimilarity were similar among species groups in Fig. 3, with difference mainly in regression slope (which is natural because you are examine species groups with great differences). Fig. S6 is very interesting, and revealed that both the taxonomic and phylogenetic components of turnover and total dissimilarity increased with longer distance. Meanwhile, these two components also showed consistent humped pattern for nestedness dissimilarity. And these patterns showed good consistence among species groups. The only exception was functional beta diversity, which differed markedly among species groups. But this is not surprising, because you used very different functional traits for different species, which may respond very

different to environmental gradients. Thus, it seems at least to me, that your results revealed some general mechanisms underlying altitudinal diversity patterns, though different diversity components and species groups are predicted to show some difference.

Response: *We thanks so much for your constructive comments. In the former edition of manuscript, we didn't summarize the general mechanisms underlying elevational beta-diversity. Unfortunately, we have paid more attention to describe all of the patterns we have detected. Following your suggestions, we substantially revised this manuscript and paid more attention to the general mechanism across different species groups. The details of revisions and corrections were in the "tracked changes" version of the manuscript.*

Reviewer's comments: The above-mentioned conclusion of the abstract seemed to be mainly based on results such as Tables 1 and Figs. 2 and S5, which compared data of the whole altitudinal gradients together. However, such overall analyses may be misleading. For instance, Fig. S5 showed that random process contributed to large proportion to altitudinal beta diversity patterns, while the effects of two niche-based processes was very weak in many cases. Considering you are examined an altitudinal gradient that is cold enough (2000-4500 m), this result is not easily believed. However, Fig S6 showed that this may be caused by the fact that there is a transition of the dominant role from environmental filtering to interspecific competition with increasing distance in many cases. When the effect of environmental filtering and interspecific competition were similar, the observed beta diversity can be not significantly different from null model expectations, which may not necessarily mean it was a result of neutral processes.

Response: *As mentioned in the last response, according to your suggestions, we have paid more attention to the general mechanisms rather than overall analyses. And we have updated the part of discussion on the underlying mechanisms. All details were in the sections of "discussion" (Lines 395-416) and "conclusions" (Lines 419-440).*

Reviewer's comments: Consequently, it may be better for you to report the 95% confidence lines of null models in comparison with observed patterns, which may provide more details of how dominant processes changes with distance or elevation.

Response: *Thanks very much for providing an efficient approach to examine “when and how the observed beta-diversity deviate from the expectations under null models”. According to your comment, we have reported the upper and lower boundaries of 95% confidence for observed beta-diversity under null model. Details could be found in Fig. 3 (Lines 704-712).*

Reviewer's comments: Thus, results such as Fig. S5 and 3 can provided interesting patterns and explanations, compared with other results. In the present MS, you have focused on other results, which is not interesting enough and made the results and discussions not very informative.

Response: *According to reviewer's comment, we have paid more attention to the results of standardized beta-diversity in the sections of discussion and conclusions. All details were in the sections of “discussion” (Lines 395-416) and “conclusions” (Lines 419-440).*

Reviewer's comments: At the same time, it is well known that beta diversity patterns are strongly affected by species numbers. And comparison of observed patterns with null model results is an effective way to distinguish the real beta diversity pattern. However, the present MS is mainly not based on these results, and have only a few discussion on the related results.

Response: *As above-responded, we have paid more attention to the results of standardized beta-diversity and interpreted the patterns detected from the accumulative bar plot and linear patterns. Details were in Lines 395-416.*

Reviewer's comments: 3) The introduction is generally well written, especially Fig. 1 (but may need some more explanation on Fig. 1 A and B for readers not familiar with beta diversity to better understand).

Response: *According to this comment, we provided more explanations on Fig. 1 A. Details were in Lines 71-79. In addition, in order to clarify the approach of decomposing, we revised the subfigure (B) in Fig.1. And we provided a concise interpretation for it in the section of “background”. A more detailed description of beta-diversity decomposition was in the appendices (Appendix file 05).*

Reviewer’s comments: But the Results and Discussions are not interesting enough and somehow too long (especially some sections of the Discussions), probably because they did not focused on some interesting patterns, such as those mentioned above.

Response: *As you suggested, we revised the section of discussion with a more attention to general patterns and mechanisms across species groups. Details could be found in the “tracked changes” version of the manuscript. In addition, we also paid more attention to the part of mechanistic estimations based on standardized beta-diversity (Lines 395-416).*

Reviewer’s comments: The methods to decompose total beta diversity into turnover and nestedness components need to be introduced briefly. Though it has been described in literatures, but this is a key method of the MS and thus need to be introduced for readers to better understand.

Response: *Following your suggestions, we provided a detailed information about the Baselga’s approach of beta-diversity decomposing approach. However, due to strict word limit in PRSB, the part of “Observed and standardized measurements of pairwise beta-diversity” was provided as a supplementary material (Appendix file 05).*

Reviewer #2:

Reviewer’s comments: This manuscript by Du et al describes patterns of beta diversity across an elevational gradient in Tibet. The topic is really interesting, and this might be a good dataset to tackle various topics in beta diversity research. At this

stage, however, the manuscript is not ready for publication. It is not clear what the actual goal of the manuscript is, the methods are lacking in important detail, and there is almost no discussion of the ecology of these various animal groups that would help the authors and the reader to understand the patterns that are described.

Response: *We thank a lot for your valuable comments and suggestions on this manuscript. According to your comments, we made substantial revisions in each section of this manuscript. The details of revision were in the following responses as well as the “tracked changes” version of the manuscript.*

Reviewer’s comments: I have made a range of suggestions below. The biggest task should be to better describe to the reader what has been found before, and what this paper will do to push the field forward. Perhaps the authors already know this, but it does not come across in this version of the manuscript.

Response: *According to your comments, as above-mentioned, we substantially modified this manuscript, line by line. In the section of Background, we have stated the concise developing history of beta-diversity including the defections of traditional species beta-diversity and the essential of integrating multiple dimensions of beta-diversity (the first paragraph), the necessities of decomposing beta-diversity (the second paragraph), and the potential mechanisms structuring beta-diversity (the third paragraph). And then we clarified our major aims in this work in the last paragraph of background (Lines 97-98).*

We agree with the comment that “The biggest task should be to better describe to the reader what has been found before, and what this paper will do to push the field forward”. Unfortunately, the patterns as well as the mechanistic inference in previous empirical studies on elevational beta-diversity were not completely consistent due to the variations in research interests, scales, taxa, and ecosystems. This was also the major reason that we attempted to reveal the general points via a multi-faceted comparative study.

Reviewer’s comments: Introduction: The introduction touches on many ideas, but

doesn't really explain or expand on any of them particularly well. Most importantly, it isn't clear exactly why the authors are interested in the topic, or what their analyses will add. What gap will the authors fill?

Response: *According to your comments, we revised the section of Background carefully. We described three parts of the development in beta-diversity: the defections of traditional species beta-diversity and the essential of integrating multiple dimensions of beta-diversity (the first paragraph), the necessities of decomposing beta-diversity (the second paragraph), and the potential mechanisms structuring beta-diversity (the third paragraph). We looked back to these facets of beta-diversity, as we attempt to understand the patterns and causes of elevational beta-diversity by integrating these thoughts under a multi-faceted comparative perspective. In the last paragraph of background, we clarified our major aims in this work (Lines 97-98). All of the details of corrections and revisions were in the "tracked changes" version of manuscript.*

Besides, in order to link the background and our research aims, as you suggested, we proposed a framework of multi-facet comparison to clarify the potential patterns and relevant mechanisms. Details were in the Table 1 (Lines 713-715).

Reviewer's comments: For example, the authors jump around from taxonomic/phylogenetic/functional perspectives on beta diversity, to beta diversity decomposition, to stochastic vs deterministic processes. These are all important topics, but the reader isn't told why they should care. In addition, the authors make many broad statements (with references), but don't unpack what this previous work has found. To my mind, it seems that the central question is really about seeing how consistent all of these different beta diversity patterns (functional vs phylogenetic, or nestedness vs turnover, or stochastic vs deterministic) are across different groups of organisms.

Response: *As above-responded, we made substantial revision in the section of Background and clarified the major aims in this work (Lines 97-98). In addition, according to your suggestions, we have proposed a framework to summarize potential*

patterns and relevant theoretical basis in the comparative analyses. The details were provided in the Table 1 (Lines 713-715).

Reviewer's comments: Have other researchers looked at this? What did they find?

Response: *Given that each topic mentioned in Background was important in community ecology and has derived hundreds of empirical studies, we failed to provide detailed findings due to word limit. Alternatively, we summarized some of the general findings. For example, as “taxonomic classification alone cannot account for ecological and evolutionary differentiation” (Lines 55-56), it was important to integrate “taxonomic, phylogenetic, and trait-based dimensions to understand the mechanisms determining community composition over space and time” (Lines 66-68); On the topic of beta-diversity decomposing, “these two antithetical processes (turnover and nestedness processes) are often mixed together, implying the necessity of beta-diversity decomposing” (Lines 75-76). In the mechanistic estimation, “the influence of both neutral and deterministic processes is widely accepted, but their relative importance typically varies across scales, environmental gradients and taxa” (Lines 92-95).*

Reviewer's comments: With this in mind, maybe it would be worth restructuring the introduction. Some more explanation and description of the wide range of approaches and results would help. Then the reader will understand that the goal in this manuscript is to try and test whether there are any consistent signals in this diversity of results. Can some key example studies be described which find conflicting findings in relation to these beta diversity topics?

Response: *According to this comment of Reviewer#2, we have modified the section of background by clarifying the major aim in this work (Lines 97-98). Then, as you suggested in the next comment, we proposed a series of predictions and their relevant mechanisms to describe the potential scenarios in the three-step of comparisons. All details were in the Table 1 (Lines 713-715).*

Reviewer's comments: Another suggestion: what do the authors predict will happen? Do they expect that functional beta will change much more than phylogenetic beta? Or that some organisms will show less beta than others? If so, why? This will help to add this missing context that I am talking about. These predictions could be discussed toward the end of this section, after the questions are outlined.

Response: *As responded above, we adopted your suggestion to propose a series of hypotheses summarizing the potential patterns and relevant mechanisms. Details were in the Table 1 (Lines 713-715).*

Reviewer's comments: Methods: There needs to be more detail about the sampling methods in the main text. Over what timescale were these data collected? Were they the same sites? Were they active or passive sampling methods? Who validated the identifications? We all three taxa sampled across the entire gradient?

Response: *In former version of the manuscript, we did not provide the information of data collection in "M&M" due to word limit. According to the comments of editor and reviewers, we provided a concise description of data collection in the main text (Lines 133-155). A more detailed description on data collection was in the supplementary materials (Appendix file 01).*

In the field surveys, the data of community composition of three animal groups (passerines, rodents, and ants) was collection under different timescales. Passerines were surveyed from 17 local sampling transects dating from June 2018/2019 to June 2020. Using snap-trapping and field observation, rodents were sampled from 9 sampling sites along elevational gradient. We conducted two snap-trapping surveys during the early (March-July) and late (July-September) wet seasons in 2014. Field observations were conducted three times at the same sites of snap-trapping, of which the first and second ones were accomplished during snap-trapping and the third field observation was carried out in August 2018. The data of ants was collected from 11 sampling sites dating from July to August 2009.

Field surveys on passerines and ants were conducted using the active approach of sampling. Field surveys on passerines were conducted using field observation and

*repeated along the same sampling transect at each sampling site. Field work on ants were accomplished using active sample collections within five 1*1 m quadrats. The field surveys on rodents were conducted using both passive (snap-trapping) and active samplings (field observation). Snap-trapping and field observation on rodents at each sampling site were repeated with a distance of at least 2,000 m.*

Species identifications for passerines were accomplished by Associate professor Liqing Fan. For rodents, Prof. Yong Ma and Dr. Zhixin Wen validated the identifications. And Prof. Zhenghui Xu contributed to the species identification of ants.

Considering the ecological characters of species groups and experimental feasibility, field surveys on three animal groups were conducted within different elevational ranges: Passerines were surveyed from 2,000 m to 4,500 m; Rodents were sampled from 2,000 m to 4,400 m; and ants were collected from 2,000 m to 4,250 m. We have tried to collect ants at 4,548 m, but we did not find any ant species. The detailed geographical information of sampling sites could be found in the dataset of community composition shared on Dryad (doi:10.5061/dryad.k98sf7m50) and Fig. S1 (Appendix file 06). The detailed information on data collection was in the Appendix file 01.

Reviewer's comments: Birds: In the supplementary material, the bird sampling section is not detailed enough. What sampling method was used? This is basic information that needs to be in the manuscript.

Response: *Field surveys on passerines were accomplished using field observation with a secondary lab identification with limited records of photos and birdsongs. As above-responded, we provided a concise sampling information in the main text in Lines 133-155. A more detailed information of data collection were provided in the Appendix file 01.*

Reviewer's comments: Rodents: What time scale did this take place over? At one point it seems to be only 2014, but later the text refers to samples from 2018? Were

the rodents released or killed? This has important implications for the findings.

Response: *Field surveys on rodents were conducted using snap-trapping and field observations. Snap-trappings at each sampling site were conducted twice during early (March-July) and late (July-September) wet seasons in 2014. In most of cases, the captured rodents were already dead when we checked the traps. If not, the animal was executed painlessly by breaking neck. Each carcass of rodent was measured in the field. The skulls and muscle tissues were taken back to the lab for further identification. Field observations were conducted for three times, of which the first and second field observations were carried out during snap-trappings, and the third time was conducted in August 2018. The details of field surveys on rodents were in the Appendix file 01.*

Reviewer's comments: Ants: So were the ants actively sampled? Do you think active sampling is a good representation of the relative abundances of the ant community? I am unsure - I think that this method leads to bias as the searcher is drawn to large, colourful or distinctive species. The collection needs to be described in detail in this paper, the authors should not just refer to a citation.

Response: *We admitted the bias of active sampling approach in the field surveys on ants. This problem has frequently troubled us in the field surveys. The active and passive samplings were both applied in the field surveys. However, in the study area in this work, it's extremely rainy especially in summer and autumn (the time when we conducted field surveys). This situation resulted in frequent failure in the passive trapping. Alternatively, under extremely bad weather, we conducted active samplings with extensive sampling efforts. We admitted this bias and its influence to community structure (especially for rare species). However, with extensive sampling efforts, the influence of uncomplete sampling should be acceptable in beta-diversity analyses [1]. After all, this situation was uncontrollable in the field.*

Reference:

1. Roden, V.J., Kocsis, Á.T., Zúschin, M. and Kiessling, W. (2018), *Reliable estimates of beta diversity with incomplete sampling. Ecology, 99: 1051-1062.*

<https://doi.org/10.1002/ecy.2201>

Reviewer's comments: In addition, the ants were sampled in a "50*50" but no units are given. Metres?

Response: *Sorry for this silly mistake. Yes, it is 'meter'. We have corrected it (Line 58 of Appendix file 01). Thanks a lot.*

Reviewer's comments: Elevational distribution interpolation: I imagine that this will have a major influence on any analysis that attempts to understand stochasticity. This procedure deterministically generates a distribution for a species that might not actually be true. It is unclear to me why this is being done in this study, why not just use the observed compositions at each elevation for each animal group? This needs to be discussed.

Response: *We totally agree with your comment "this will have a major influence on any analysis that attempts to understand stochasticity". The procedure of elevational distribution interpolation will artificially enlarge the proportion of common species across assemblages, which will reduce the degree of dissimilarity and enlarge the effect of stochasticity. As you suggested, we reran all of the analyses only based on the raw presence-absence community data. According to the latest analytical results, this manuscript has been totally revised from the beginning to the end. The details of the revisions and corrections were in the "tracked changes" version of manuscript.*

Reviewer's comments: I do not think that measuring size for each taxon is enough to be called a functional diversity analysis. Functional traits and functional diversity, as written in the introduction, should capture something of the ecological niche of the organism. Using only size, and nothing else, is poor. I would advise reading some of the seminal functional trait papers such as:

McGill, B.J., Enquist, B.J., Weiher, E. and Westoby, M., 2006. Rebuilding community ecology from functional traits. *Trends in ecology & evolution*, 21(4), pp.178-185.

Cadotte, M.W., Carscadden, K. and Mirotnick, N., 2011. Beyond species:

functional diversity and the maintenance of ecological processes and services. *Journal of applied ecology*, 48(5), pp.1079-1087.

Response: *In this manuscript, the size-related morphologies were used in the trait-based beta-diversity analyses for two major reasons. In this work, we aimed to reveal the general patterns and mechanisms in elevational beta-diversities under a “comparative perspective”. This work was extraordinary challenging especially at the trait-based dimension. Well known, the three animal groups (passerines, rodents, and ants) displayed distinct roles in ecosystems due to their differences in ecological characters. Hence, we needed to choose “comparable” functional attributes displaying common or similar functional roles in the niche space. Body size is one of the most important ecological characters affecting animal interspecific competition and resource access [2-4]. As size-related morphologies display similar ecological roles for distinct animal groups during local adaptation, they have been widely used in the meta-analyses of animal beta-diversity across species groups and ecosystems [5-7]. On the other hand, the question of “how a certain animal group adapts to local environmental gradient via functional attributes” was out of our research scopes in the current manuscript. Instead, we attempted to answer “how the dissimilarity of body size varies across animal communities along elevational gradient” and “whether the patterns were consistent across animal groups”.*

In addition, as we only used the size-related morphological attributes in functional analyses, we revised the “functional analyses” as “trait-based analyses”. Details were in the “tracked changes” version of the manuscript.

Reference:

2. Blackburn, T.M., Brown, V.K., Doube, B.M., Greenwood, J.J.D., Lawton, J.H. & Stork, N.E. 1993 *The Relationship Between Abundance and Body-Size in Natural Animal Assemblages*. *Journal of Animal Ecology* 62, 519-528. (doi:10.2307/5201).
3. Blackburn, T.M. & Gaston, K.J. 1994 *Animal body size distributions: patterns, mechanisms and implications*. *Trends Ecol Evol* 9, 471-474. (doi:10.1016/0169-5347(94)90311-5).

4. Lafferty, K.D. & Kuris, A.M. 2002 Trophic strategies, animal diversity and body size. *Trends in Ecology & Evolution* **17**, 507-513. (doi:Pii S0169-5347(02)02615-0 (Doi 10.1016/S0169-5347(02)02615-0).
5. Soininen, J., Lennon, J.J. & Hillebrand, H. 2007 A multivariate analysis of beta diversity across organisms and environments. *Ecology* **88**, 2830-2838. (doi:10.1890/06-1730.1).
6. Soininen, J., Heino, J. & Wang, J.J. 2018 A meta-analysis of nestedness and turnover components of beta diversity across organisms and ecosystems. *Global Ecology and Biogeography* **27**, 96-109. (doi:10.1111/geb.12660).
7. McGill, B.J., Enquist, B.J., Weiher, E. & Westoby, M. 2006 Rebuilding community ecology from functional traits. *Trends Ecol Evol* **21**, 178-185. (doi:10.1016/j.tree.2006.02.002).

Reviewer's comments: Finally, it is not actually stated what constitutes a "community" in this context. Is it the sampling sites? Is it every 100 m of elevation? This is critical information that is needed to assess the manuscript.

Response: *In the current version of manuscript, we have stated the ecological meanings of the concepts of "community" and "assemblage" that "all species occurring in each sampling site". Details were in Lines 218-219 in the section of "M&M".*

Reviewer's comments: Results: The results are presented OK, but I cannot assess how valid they are given the lack of detail in the methods sections.

Response: *As above-responded above, we have rerun all of the statistical analyses based on the raw presence-absence community data. According to the latest analytical results, we substantially revised this manuscript. Details were in the "tracked changes" version of the manuscript.*

Reviewer's comments: I wondered why there were no plots of the SES analyses?

Response: *In former version of the manuscript, we did not illustrate the plots in the*

SES analyses due to scatter overlap which will blur the fitting lines. According to this comment, we have revised the scatter plot of the standardized beta-diversity. Detailed revisions were in Fig. S5 (Lines 22-27 in Appendix file 06).

Reviewer's comments: What is clearly missing, however, is discussion of any ecology. What do these various species and animal groups do? What traits do they have that might drive various beta diversity patterns? Unfortunately, this is probably where the focus on only size related traits will let the manuscript down.

Response: *According the comments of both editor and reviewers, we carefully revised this manuscript. In the discussion section, we paid more attention to the consistence across species groups. Details could be found in the section of discussion.*

By using the size-related morphologies, we paid more attention to examine whether species turnover or nestedness process occurs more often in size-similar or size-different lineages (the second level of comparison). Our results supported the former that species turnover and nestedness often occurred among phylogenetic relative exhibiting similar trait performance (similar body size). The ecological roles of size-related morphologies were not deeply discussed due to their inconsistent influence in the life history of different species groups. Details were in Lines 371-394.

Reviewer's comments: The lack of a clear goal in the introduction also lets this discussion section down. What is the overarching question? The discussion talks a lot about various topics, but it is not clear how they fit together. Do they differences between birds, rodents and ants have anything to say about these patterns? Are these patterns expected or surprising? I suggest that more thought is put into the introduction and the framing of the questions, then this section will be able to flow naturally from that.

Response: *As responded before, we have conducted a substantial revision in each section of this manuscript according to the comments of both editor and reviewers. In the current version of the manuscript, the the major aims in this work have been clarified (Lines 97-98). In addition, following your suggestions, we have proposed a*

framework to summarize potential scenarios in the comparative analyses. Details were in the Table 1 (Lines 713-715).

Reviewer's comments: Line 26-28: "The domain process..." I'm not sure what this sentence is saying, can it be rephrased?

Response: *According to your comment, this sentence has been rephrased (Lines 28-29).*

Reviewer's comments: Line 28: "Second", but where is the "first" in this abstract?

Response: *This mistake has been corrected in current version of manuscript (Lines 29-31).*

Reviewer's comments: Line 31-32: "the rates of dissimilarities". I'm not sure what this means. The rate of decay of dissimilarity/similarity with elevational distance? Again, perhaps rephrase.

Response: *The phrase "the rates of dissimilarities" has been corrected as "the rate of increase in species/phylogenetic dissimilarity" (Lines 35-38).*

Reviewer's comments: Line 34: "leading process". I have no idea on this, again, perhaps it is best to make the abstract slightly less technical.

Response: *According to this comment, the phrase "leading process" has been corrected (Line 34).*

Reviewer's comments: Line 47: "species assembly". I agree, but I think this concept needs to be introduced otherwise it will probably be unclear to the large number of readers of Proceedings B who are not community or macro-ecologists.

Response: *The phrase "species assembly" has been corrected as "community structure" (Lines 50-51).*

Reviewer's comments: Line 56-57: "local and regional processes", sure, but how?

This sounds like (and is) an important statement, but it isn't explained properly. I think that concepts like this need to be explained in full if the reader is to understand what the paper is trying to do. Maybe it is unnecessary to mention for the goal at hand?

Response: *The phrase “local and regional processes” here indicates the local and regional processes affecting the patterns of beta-diversity. This phrase was cited from the Graham’s research paper on phylogenetic beta-diversity [8]. The local processes mainly focus on the local ecological process (i.e., local climate, habitat heterogeneity, and local topography) at fine spatio-temporal scales. The regional processes involve regional histories in geology (i.e., geological movement), climate (i.e., glacial-interglacial cycle), lineage evolution and other processes at broad spatio-temporal scales. How these processes together affect current patterns of beta-diversity was extremely complicated and often varied across research systems. Here, we quoted this inference with an aim to clarify the importance of phylogenetic and trait-based beta-diversities. Therefore, we did not expand this point in the background section of this manuscript. Alternatively, relevant interpretations were partially presented in the theoretical framework in the Table 1(Lines 713-715).*

Reference:

8. Graham, C.H. & Fine, P.V. 2008 Phylogenetic beta diversity: linking ecological and evolutionary processes across space in time. *Ecol Lett* 11, 1265-1277. (doi:10.1111/j.1461-0248.2008.01256.x).

Reviewer’s comments: Line 67: "origin of dissimilarity". I find this a strange phrase. Is it a quote?

Response: *This phrase “origin of dissimilarity” has been corrected (Lines 69-70).*

Reviewer’s comments: Line 95: "rate of dissimilarity". Again, rate relative to what? Change in elevation? Make this clear for the reader.

Response: *The phrase "rate of dissimilarity" has been revised as “the rate of increase in dissimilarity”. Similar corrections could be found in Lines 231, 270, 278, and 313.*

Reviewer's comments: Line 107: "quantified Lastly". I think there is some text missing here...

Response: *This sentence has been removed in current version of manuscript.*

Reviewer's comments: Line 121: "hungriness". I don't now what this means.

Response: *As a writing error, the word "hungriness" has been corrected as "tundra and desert" (Line 131).*

Reviewer's comments: Line 127: I do not think it is appropriate to put all of the sampling information into the appendix. This needs to be described, at least in brief, in the main text. This is my view, perhaps the editors will disagree.

Response: *According to your suggestions, we provided a concise description of data collection in the main text. Due to the word limit, a more detailed sampling information was in Appendix file 01.*

Reviewer's comments: Line 154: The mean of each measure, or a mean across the 5? This is unclear, but the meanings could be very different.

Response: *The phrase "the mean values of five measures" has been corrected as "the mean of each measure" (Line 185).*

Reviewer's comments: Line 158: "body size" in ants. How was this measured? There are number of ways to measure body size in ant workers. This detail needs to be in the manuscript, how else could someone else read the paper and replicate the study?!

Response: *The "body size" of ants were measured using the total natural length of head, thoracic segments, and abdominal segments (Lines 191-192).*

Reviewer's comments: Line 162: So size is the only functional trait for each species? Do the authors think that size alone captures the ecological niche? What about species that are similar in size but have different diets or habitat preferences that other traits

might reveal?

Response: *As responded before, our major aims in this work are to reveal the general patterns and mechanisms of beta-diversity. By using size-related morphological traits, trait-based beta-diversity in this work is aiming to examine whether the size-related traits exhibited similar or consistent patterns across species groups. We admit that diet, behaviours, and other ecological characters display important roles in species life history. But the question of “how the niche space of focal species group varied along elevational gradient” was out of the scope in current manuscript. Alternatively, we only focused on the size-related morphology as this ecological character were considering as comparable across animal groups.*

Reviewer’s comments: Line 163: Why just the first two components?

Response: *We used the first two components in principal component analysis (PCA) to represent the variations in size-related morphologies. This was because that the first two components accounted for 70%-100% of the total trait variations (Appendix file 03). And the standard deviations of the first and second components are higher than one or approaching to one, indicating their importance in the total trait variations (not presented in the manuscript).*

Reviewer’s comments: Line 171-174: Yes, but you are using a dendrogram method to calculate functional beta diversity, so surely this is not relevant?

Response: *In the convex hull-based approach, the functional or niche position of certain community is measured using convex hull, which needs at least three points (species) in the function or niche space. However, in our dataset, there are many assemblages only containing two species, especially for rodents and ants. That’s the reason why we used a dendrogram method to calculate trait-based beta-diversity.*

Reviewer’s comments: Line 177-180: This is all fine, but I think it needs some more explanation for those readers who are not experts in beta diversity decomposition.

What exactly does this decomposition do? Is it a standard technique? (I would say that

yes it is a standard technique now, but you need to let the reader know this and provide context).

Response: *Following your suggestion, we provided a more detailed information of the approach of beta-diversity decomposition. However, due strict word limit, this part was provided as supplementary material (Appendix file 05).*

Reviewer's comments: Line 182: I'm not sure if those studies "proposed" that null model. You should find the paper or papers that originally proposed and used it. You should also describe it in more detail in this manuscript - the next few lines do not adequately explain how this procedure "controls for species richness".

Response: *According to this comment, we quoted the seed paper of relevant null model (Line 29, Appendix file 05). And we have provided a more detailed information about the procedure of randomization. Details were in Lines 29-34 in the Appendix file 05.*

Reviewer's comments: Line 205: It isn't really clear to me what "components" and "facets" means in this context.

Response: *The word "components" in this manuscript indicates "the turnover and nestedness-resultant compositional components" of beta-diversity, whereas "facets" mean the "dimensions (species, phylogenetic, and trait-based dimensions)" of beta-diversity. Besides, this sentence has been revised in current version (Lines 227-229).*

Reviewer's comments: Line 210-213: Why use regression to test beta diversity against elevational distance, but then a mantel test to test the relationships among beta diversity types? This needs to be explained.

Response: *The part of inter-dimensional correlation analyses was removed out of the current version of the manuscript, as it was not relevant to our major aims. Besides, according to reviewer's comment, we used a unified linear regression model to examine the elevational patterns of both observed and standardized beta-diversities with elevational distance. Details were in Lines 229-231 and Lines 248-250.*

Reviewer's comments: Line 267-269: AICs are not a measure of model fit, and they cannot be used to compare entirely different models sets (i.e. models of turnover vs models of nestedness).

Response: *We totally agree with the comment "AICs are not a measure of model fit, and they cannot be used to compare entirely different models sets". In previous version of manuscript, by checking the AICs in linear and binomial models, we compared the degrees of model fitting for each beta-diversity. Probably, our misleading literal expression has led this problem. As above-responded, a unified linear model was used to examine the patterns of both observed and standardized beta-diversity with elevational distance. Details were in Lines 229-231 and Lines 248-250.*

Reviewer's comments: Figure 3: I presume the x-axis is elevational distance? Then why is it log transformed? I don't see why this would be necessary. This is really unclear.

Response: *According to your comment, the linear patterns of beta-diversity with elevational distance were examined without log-transformation in elevations. Details were in Lines 229-231 and Lines 248-250. The results of linear regressions for observed beta-diversities were in Fig.3 (Lines 704-711), and Table S2 (Lines 34-36 in Appendix file 06). The results of linear regressions for standardized beta-diversities could be found in Fig. S5 (Lines 22-27 of Appendix file 06), and Table S4 (Lines 43-47 in Appendix file 06).*

Reviewer's comments: Line 290: Is "analogs" the right word?

Response: *The word "analogs" has been corrected as "measures" in Line 297.*

Reviewer's comments: Line 322: Do you mean "the main" rather than "domain"?

Response: *Yes, and we have revised this question as "What dominates beta-diversity, turnover or nestedness?" (Line 318).*

Reviewer's comments: Line 353-358: Yes, sure. Is that what you found? You need to link this discussion back to your actual results.

Response: *According to the comments of editor and reviewers, the section of discussion has been substantially revised. We quoted these inferences in previous empirical studies as these findings were consistent with those detected in this work (Lines 359-361). Besides, these findings in comparative analyses were corresponding to the hypotheses or alternative hypotheses proposed in our theoretical framework (Table 1) (Lines 713-715).*

Thanks very much for these valuable comments from Associate editor, Reviewer #1, and Reviewer #2. These constructive comments are helpful for us to revise this paper. We hope that the Editors and Reviewers could be satisfied with the revisions made for the manuscript.

Thanks very much for your consideration.

Sincerely,

Yuanbao Du

Institute of Zoology, Chinese Academy of Sciences,

No.1, Beichen West Road, Chaoyang District

Beijing, China

2020-12-09

Appendix B

Responding to Editor's and Reviewer's comments

Reviewer #1:

Reviewer's comments: This manuscript (MS) has been substantially revised, especially in the way to analyze data and to interpret the results. I am glad to see the authors pay more attention to summarize general patterns from their results, which is an important aim for us to compare the patterns from different species groups and diversity components and dimensions. While the MS has been clearly improved, there are still some points that needed to be clarified, or improved, as summarized below.

Response: *We deeply appreciate your professional and constructive comments and suggestions, which were definitely helpful for us to improve and revise the structure as well as the details of the MS. According to these comments and suggestions, we revised the former edition of MS line by line. The details of revision can be found in the "tracked changes" version of MS and in the following responses.*

Reviewer's comments: Major concerns: 1) L33: In abstract, you stated that: "deterministic and neutral processes have jointly contributed to driving community beta-diversity, ...". However, as mentioned last time, the large proportion of contribution of the so-called random processes in Fig. S4 may simply be a result of transition from environmental filtering to negative interspecific effect, which causes a large proportion around the middle elevational distance not distinguishable from null model predictions (this can be somehow indicated from the new Fig. 3, though Fig. 3 now included too many lines and some patterns were not clear. I would suggest you to split Fig. 3 into several ones and move Fig. 3 into appendices, because it seems that you did not mention Fig. 3 in the new MS and the related results were based on Fig. S5). It is nice that you have discussed about this point at the end of Discussion section. Thus it is not appropriate for you to conclude the above-mentioned sentence in the Abstract.

Response: *We agree with you that it is not appropriate to conclude "deterministic and neutral processes have jointly contributed to driving community beta-diversity, ...".*

Because, as you mentioned, we only inferred the dynamics of opposite deterministic processes (e.g., environmental filtering and competitive exclusion) in driving elevational beta-diversity. Based on this, we revised all of the related contents throughout the MS. Additionally, we followed your suggestion to split Fig. 3 and move it into the appendices (the Fig. S4 in current MS). Because of the importance of the former Fig. S5 in explaining the dynamics of niche processes, it is now presented as the Fig. 3 in the current MS. All the details can be found in the related sections in the “tracked changes” version of main text as well as related appendices.

Reviewer’s comments: Also, in the middle panel of Fig. 1F (and the related words in texts), this point should also be clarified (e.g. L247 is nice, but L117 and L80-94 may lead some readers to interpret your Fig. S4 as largely caused by neutral processes). I believe this is important, because your results till now cannot prove that neutral processes played a so important role as suggested in Fig. S4. For instance, in most part of the Discussions you explained your elevational patterns with deterministic processes (e.g. L356-363 are well written). These explanations based on deterministic processes are in contradiction with the high contribution of neutral processes as suggested in Fig. S4. So, I believe it is better for you to revise related word throughout the MS, or, I am not sure if you really need Fig. S4.

Response: *We agree with this comment. Given the former Fig. S4 was not actually helpful to disentangle the effect of neutral and niche-based processes, we removed it from the supplementary materials as you suggested. The only point referred from the analyses in standardized beta-diversity is the transition of opposite deterministic process as elevational distance increased. As corresponding to this finding, we edited the related content in the introduction, discussion, and conclusion in this new version of the MS. Details are clearly presented in the “tracked changes” version of main text and related appendices.*

Reviewer’s comments: 2) The MS has provided many results in the appendices, but it seems that not all of them were necessary, which made the MS complicated. It also

seems that some of the results were not mentioned in the Discussions at all. Please consider to remain only the most important ones.

Response: *As we addressed above, following your suggestion, we removed the Fig. S4 from the appendices. Moreover, to better explore the results shown in the Table S3, we included a new paragraph in the discussion to proper address the importance of environments effects on the diversity patterns.*

Reviewer's comments: Meanwhile, the connections between your results and discussions are not clear. For instance, L384 said that “Our analysis on the best environmental predictors supports this”. However, how it was supported was not clearly explained, and it is easily to made the readers confused. Frankly, I did not find interesting discussions about the results in Table S3. It's good and important to relate diversity patterns to climate factors, in addition to elevational gradients. But without good discussions and interpretations, Table S3 seems not necessary. For instance, why NPP and temperature is more important in explaining beta diversity patterns than precipitation and humidity? This is an interesting result, but was not well explained.

Response: *We totally agree with you about the importance of better discuss how the diversity patterns relate to climate factors. As an evidence for the role of niche-based process (species-environment interaction) in driving beta-diversity, the results of best environmental predictors for observed beta-diversity measures are helpful for us to understand the patterns of multi-dimensional comparisons. Therefore, we provided a new paragraph with a concise discussion on the environmental predictor effects (please see **Lines 391-402**).*

Reviewer's comments: Generally speaking, the Introduction and methods sections are well written now, but the Results and Discussion sections are not well organized and sometimes hard to read. Please revise so that the MS can be clear in expressing your logics.

Response: *According to your comments, we have conducted a substantial revision throughout this MS, especially in the related contents of Abstract, Results, Discussion,*

and Conclusions. Details can be found in the “tracked changes” version of the MS.

Reviewer’s comments: Minor points: L25: remain poorly understood. This is clearly not appropriate.

Response: *The statement of “remain poorly understood” has been corrected as “multi-dimensional comparative studies remain scarce” (Line 27).*

Reviewer’s comments: L30: has dominated species beta-diversity -> has dominated altitudinal patterns of species beta-diversity in your study.

Response: *We edited this sentence following the reviewer’s suggestion (Lines 31-32).*

Reviewer’s comments: L56: cannot account for ecological and evolutionary differentiation-> cannot account for functional and evolutionary differentiation

Response: *We have modified this sentence as you suggested (Line 59).*

Reviewer #3:

Reviewer’s comments: As a new referee, I read carefully this resubmitted MS. While the authors have done a very good revision work for several issues, I have major concerns about the use of functional dendrogram to calculate functional beta diversity.

Response: *We deeply thank you for the constructive and technical comments on the part of calculating functional beta-diversity. As an indivisible part of multi-dimensional beta-diversity, the patterns related to functional beta-diversity greatly affect the final inference on the underlying mechanisms of beta-diversity in this MS. According to your comments, we recalculated the observed and standardized trait beta-diversity measures using the convex hull approach proposed by Villéger et al. (2013), and rerun all related analyses. Details can be found in the “tracked changes” version of the MS, related appendices, and in the following responses.*

Reference:

Villéger, S., Grenouillet, G. & Brosse, S. 2013 Decomposing functional β -diversity

reveals that low functional β -diversity is driven by low functional turnover in European fish assemblages. Glob Ecol Biogeogr **22**, 671-681. (doi:10.1111/geb.12021).

Reviewer's comments: I am concern that the authors did not provide a quantitative assessment of how functional distances are faithfully represented in the functional dendrogram, while recent studies showed that this represents an important step with strong consequences when quantifying functional diversity (see Maire et al. 2015). The authors should show the relationship between the distances derived from the functional dendrogram and the original Euclidean distance matrix (one might expect a strong triangular relationship with closely related species being far apart in the functional dendrogram) and (ii) used a metric, i.e. not the correlation coefficient that is biased but the Norm2 of Merigot et al. (2010) in Ecology or mSD of Maire et al. (2015) in Global Ecology and Biogeography.

Response: *We agree with your comment that the methodological flaw of the dendrogram-based approach for calculating functional beta-diversity might lead to erroneous results in the observed patterns and in predicting underlying mechanism in this MS. Admittedly, in the former version of MS, we did realize the methodological limitation of convex hull approach and the low richness in the assemblages of ant and rodent in our dataset. But we did overlook the statistical flaws of dendrogram-based approach. Following your suggestions, we rerun all analyses. The trait beta-diversity patterns and related mechanistic inference using convex hull approach significantly differed from those obtained from dendrogram-based approach (details to see the former and new Fig. 2 in the "tracked changes" version of MS). Therefore, in the new version of the MS, we only kept the results derived from the convex hull approach. Details can be found in the revised results and discussion of the MS.*

Reviewer's comments: The main point is that you do not only loss information using a functional dendrogram but most importantly, you provide spurious functional distances between species, with a strong triangular relationship between the

cophenetic distances derived from the functional dendrogram and the original Euclidean distances (see Fig 3 in Maire et al. 2015). In contrast, using a functional space, you only lose information but this loss is limited with the increasing number of axes used to build the functional space (see again Fig 3 with the mean squared deviation, mSD decreasing with increasing number of axes). That's why it is an important step to determine the number of axes to be used to derive the FD metrics. It is a « compromise » between the quality of the representation of the original distances and the time to compute the indices, but see Maire et al. 2015 to see how to define it.

Response: *Given the significant difference in the patterns of trait beta-diversity calculated using two approaches and the methodological flaw of dendrogram-based approach described in Villeger et al. 2017, we updated all of related content according to the new results derived from the convex hull approach.*

Reference:

*Villeger, S., Maire, E. & Leprieux, F.J.E.I. 2017 On the risks of using dendrograms to measure functional diversity and multidimensional spaces to measure phylogenetic diversity: a comment on Sobral et al.(2016). Ecol lett **20**, 554-557.*

(doi:10.1111/ele.12750)

Reviewer's comments: I agree that a functional space-based approach has limitations such as samples with lower number of species than the number of selected PCA or PCOA axes cannot be used! But, at least, the authors could have calculated functional beta diversity metrics based on a functional space approach using a subset of data (i.e. only considering samples with 3 or more species), and then calculate the correlation (Mantel test using Pearson or Spearman's metrics) between the two matrices, in order to show whether the two approaches lead to similar conclusions (See Villeger et al. 2013 that showed that these two approaches can lead to contrasting conclusions or Villeger et al. 2018 in Ecology Letters).

Response: *As addressed above, given the contrasting results obtained from two approaches and obvious methodological flaws of dendrogram-based approach, we*

rerun all analyses using the convex hull approach. According to the new results, we have conducted substantial revision throughout the MS.

Reviewer's comments: In addition, the justifications made by the authors are only included in the supplementary materials, with no reference in the main text (in the methodological part).

Response: *According to this comment, we have provided a concise statement of calculating observed and standardized beta-diversities (Lines 221-235). Due to the word limit during MS submitting, the more detailed information of calculating observed and standardized beta-diversity was provided as an appendix (Appendix file 05).*

Reviewer's comments: Overall, I have strong doubt about the results obtained using a dendrogram-based approach because the authors did not demonstrate, quantitatively, that the beta-diversity metrics based on a functional space and a dendrogram, respectively, lead to similar results! If the authors can demonstrate that their functional dendrogram based approach did not affect the results and conclusions, I think that this study can represent a significant contribution to the field of large-scale community ecology.

References:

Villéger S., Maire E., Leprieur F. 2017. On the risks of using dendrograms to measure functional diversity and multidimensional spaces to measure phylogenetic diversity: a comment on Sobral et al. (2016). *Ecology Letters*. 20: 554–557.

Maire E., Grenouillet G., Brosse S. & Villéger S. 2015. How many dimensions to accurately assess functional diversity? A pragmatic approach for assessing the quality of functional spaces. *Global Ecology and Biogeography*. 24: 728–740.

Villéger S., Grenouillet G. & Brosse S. 2013. Decomposing functional β -diversity reveals that low functional β -diversity is driven by low functional turnover in European fish assemblages. *Global Ecology and Biogeography*. 22: 671-681.

Response: *We deeply appreciate your constructive comments, and hope you are*

satisfied with this new version of the MS.

We would like to express our gratitude to the Editors and Reviewers for the very constructive suggestions and comments. The comments were helpful for us to revise and improve the quality of our MS. We hope that the Editors and Reviewers are satisfied with the revisions made in the manuscript.

Thanks very much for your consideration.

Yuanbao Du

Institute of Zoology, Chinese Academy of Sciences,

No.1, Beichen West Road, Chaoyang District

Beijing, China.

Feb 9th, 2021

Appendix C

Responding to Editor's and Reviewer's comments

Reviewer #1:

Reviewer's comments: This manuscript (MS) has been improved again. I do not have further major concerns except for a few minor points listed below.

Response: Again, we deeply appreciate Reviewer #1 for the professional and constructive comments, which are definitely helpful to for us to improve and revise this MS. The details of revision have been listed as follows.

Reviewer's comments: L37~38: Maybe it is better to said something like: the effect of A increased with as elevational distance increased, while that of B decrease, which lead to ..., so that the readers can get more clear information.

Response: We have adopted Reviewer#1's suggestion and revised the sentence as “*as elevational distance increased, the contradict dynamics of environmental filtering and limiting similarity have jointly led the elevational patterns of beta-diversity, especially at taxonomic dimension*” (Lines 36-39).

Reviewer's comments: A related point is the change to the use of “limiting/promoting dissimilarity” in the new MS (e.g. Fig. 1 and many sentences elsewhere). These terms are less easily to be understood than the commonly used “environmental filtering vs. limiting similarity“, especially because limiting dissimilarity has converse meaning with the widely used “limiting similarity”. This markedly decreased the readability of some sentences (e.g. “Conversely, the decreasing promoting dissimilarity and the increasing limiting dissimilarity jointly contributed to the elevational patterns of nestedness-resultant dissimilarity”. This sentence in Discussions is hard to understand). If there's no essential difference, I would suggest to use the commonly used terms. However, this is only a suggestion.

Response: We agree with this comment. The “promoting dissimilarity” in Fig. 1 and main text have been corrected as “limiting similarity”, whereas “limiting dissimilarity” has been revised as “environmental filtering”(Lines 734 and 412-418).

Reviewer's comments: L399: “By comparison, due to the localized monsoon climate and relative high altitude in our study area, AP and PET appear relatively less important for the observed beta-diversity of three animal groups.” PET is an index for energy availability, and is similar as temperature sum. This sentence may give some readers an impression that PET is similar as AP. In you data, PET and NPP were extracted from global dataset, while AMT, AP and AMH were based on field-measured data. So it is possible that the low explanatory power of PET is because you PET data is not accurate enough compared with AMT.

Response: *We agree with the comment “it is possible that the low explanatory power of PET is because you PET data is not accurate enough compared with AMT”. Accordingly, the related sentences in discussion section have been revised (Lines 398-402).*

We deeply appreciate Associate editor and three anonymous reviewers for their kindly efforts and comments on previous editions of this MS. These comments are helpful for us to improve this MS.

Thanks very much for your consideration.

Sincerely,

Yuanbao Du

Institute of Zoology, Chinese Academy of Sciences,

No.1, Beichen West Road, Chaoyang District

Beijing, China

2021-03-22